# Snapin promotes HIV-1 transmission from dendritic cells by dampening TLR8 signaling

Elham Khatamzas[1,†], Madeleine Maria Hipp[1,†], Daniel Gaughan[1], Tica Pichulik[1], Alasdair Leslie[2], Ricardo A Fernandes[3], Daniele Muraro[1], Sarah Booth[4], Kieran Zausmer[1], Mei-Yi Sun[1], Benedikt Kessler[2], Sarah Rowland-Jones[5], Vincenzo Cerundolo[1] & Alison Simmons[1,6,*] (iD)

## Abstract

HIV-1 traffics through dendritic cells (DCs) en route to establishing a productive infection in T lymphocytes but fails to induce an innate immune response. Within DC endosomes, HIV-1 somehow evades detection by the pattern-recognition receptor (PRR) Toll-like receptor 8 (TLR8). Using a phosphoproteomic approach, we identified a robust and diverse signaling cascade triggered by HIV-1 upon entry into human DCs. A secondary siRNA screen of the identified signaling factors revealed several new mediators of HIV-1 *trans*-infection of CD4[+] T cells in DCs, including the dynein motor protein Snapin. Inhibition of Snapin enhanced localization of HIV-1 with TLR8[+] early endosomes, triggered a pro-inflammatory response, and inhibited *trans*-infection of CD4[+] T cells. Snapin inhibited TLR8 signaling in the absence of HIV-1 and is a general regulator of endosomal maturation. Thus, we identify a new mechanism of innate immune sensing by TLR8 in DCs, which is exploited by HIV-1 to promote transmission.

**Keywords** HIV-1 and dendritic cells; HIV-1 transmission; Snapin and vesicular trafficking; TLR8 sensing
**Subject Categories** Immunology; Microbiology, Virology & Host Pathogen Interaction
**The EMBO Journal** (2017) 36: 2998–3011

## Introduction

Myeloid dendritic cells (DCs) are important antigen-presenting cells that are thought to be the first cells encountered by HIV-1 in the mucosa during transmission (Borrow, 2011). HIV-1 evades detection by myeloid DCs, leading to defective cell intrinsic innate immune responses (Granelli-Piperno *et al*, 2004; Krathwohl *et al*, 2006;

Sabado *et al*, 2010; Manel *et al*, 2011; Frleta *et al*, 2012) and facilitating viral spread by promoting *trans*-infection of T lymphocytes (Menager & Littman, 2016). This helps establish a latent reservoir of HIV-1 that is resistant to anti-retroviral drugs, hampering efforts to cure the disease.

HIV-1 escapes detection in DCs during viral uptake, intracellular trafficking and in the cytosol (Luban, 2012). During uptake, the virus binds the C-type lectin DC-SIGN, which triggers a tolerogenic immune response involving downregulation of specific immune signaling molecules that may help the virus evade early detection (Hodges *et al*, 2007; Shan *et al*, 2007). Within the cytosol, HIV-1 encounters the deoxyribose nucleoside triphosphate hydrolase SAMHD1, which limits the efficiency of HIV reverse transcription and inhibits replication (Goldstone *et al*, 2011; Hrecka *et al*, 2011; Laguette *et al*, 2011). This prevents detection of reverse-transcribed viral DNA and activation of interferon responses (Gao *et al*, 2013; Bridgeman *et al*, 2015; Gentili *et al*, 2015). Other factors contributing to HIV-1 escape in DCs have been described and include TREX1, an exonuclease that degrades reverse-transcribed HIV DNA in the cytoplasm (Yan *et al*, 2010), and APOBEC3G/3F (A3G/3F) whose knockdown enhances HIV-1 infection of DCs associated with G-to-A hypermutation of the HIV genome (Pion *et al*, 2006). These observations explain how HIV-1 avoids innate sensing during uptake into DCs and in the cytosol. But it remains unknown how it evades detection within endosomes, and how internalized HIV-1 is effectively transferred to CD4[+] T cells during *trans*-infection (McDonald, 2010; Rinaldo, 2013).

It is well established that a proportion of virions reside within endosomes where pattern-recognition receptors (PRRs) such as Toll-like receptor 8 (TLR8) should recognize single-stranded RNA derived from the HIV-1 long terminal repeat (LTR) (Heil *et al*, 2004). TLR8 can detect HIV-1 in DCs, as it is required for inducing viral transcription (Gringhuis *et al*, 2010), but it somehow fails to generate robust anti-viral cytokine responses (Granelli-Piperno *et al*, 2004). TLR8 recognizes ligands

1 MRC Human Immunology Unit, Weatherall Institute of Molecular Medicine, Oxford University, Oxford, UK
2 KwaZulu-Natal Research Institute for TB & HIV, Durban, South Africa
3 Molecular and Cellular Physiology and Structural Biology, Stanford University School of Medicine, Stanford, CA, USA
4 Immunology & Immunotherapy, College of Medical & Dental Sciences, University of Birmingham, Birmingham, UK
5 Nuffield Department of Clinical Medicine, University of Oxford, NDMRB, Oxford, UK
6 Translational Gastroenterology Unit, Nuffield Department of Medicine, University of Oxford, Oxford, UK
*Corresponding author. Tel: +44 1865 222 628; E-mail: alison.simmons@imm.ox.ac.uk
†These authors contributed equally to this work

such as single-stranded RNA and imidazoquinolines triggering distinct signaling cascades (Colak *et al*, 2014) that lead to the activation of interferon-regulatory factor (IRF)-dependent interferon responses and NF-κB-dependent pro-inflammatory cytokine responses in DCs. However, the mechanisms by which TLR8 signals in DCs are poorly understood and appear to differ between mice and humans (Ohto *et al*, 2014). Recognition of viral nucleic acids by TLRs within endolysosomal compartments often requires TLR cleavage and trafficking from the endosomes to lysosome-related organelles. And there is mounting evidence that effective signaling by endosomal TLRs in DCs requires correct spatial compartmentalization to enable the juxtaposition of key components of their signaling machinery to initiate appropriate transcriptional responses (Sasai *et al*, 2010; Mantegazza *et al*, 2012). However, the molecular components required for TLR8 signaling in DCs and how HIV-1 might manipulate that to avoid interferon responses while enhancing *trans*-infection are unclear.

In this work, we used a phosphoproteomic approach and a secondary siRNA screen to characterize signaling induced by HIV-1 upon entry to immature monocyte-derived DCs and identify those signaling molecules required for *trans*-infection of autologous CD4$^+$ T cells.

# Results

## HIV-1 signaling in DCs

We took an unbiased proteomic approach to characterize the signaling response (signalosome) in DCs induced by HIV-1 entry. Immature monocyte-derived DCs were mock infected or exposed to the HIV-1 BaL strain for 10 min prior to lysis and phosphoenrichment. Ten minutes was chosen as the time point where greatest changes in phosphorylation were observed between mock and HIV-1 exposed cells during time courses of HIV-1 exposure (data not shown). Figure 1A shows changes in tyrosine phosphorylation following DC exposure to HIV-1. Phosphoenriched eluates were subjected to off-gel fractionation to reduce sample complexity and enhance detection of low abundance proteins prior to LC/MS/MS analysis (Appendix Fig S1 and schematic of experimental approach Fig 1B). Raw data obtained by LC-MS/MS was processed using the Central Proteomics Facilities Pipeline (CPFP) (Trudgian *et al*, 2010), which searches three different databases: Mascot (Matrix Science, London, UK), OMSSA (Geer *et al*, 2004), and TANDEM (Craig & Beavis, 2004), to increase both the number of protein identifications and the confidence in the data obtained. On this platform, peptide identifications from each search engine are validated with PeptideProphet (Nesvizhskii *et al*, 2003; Keller *et al*, 2005). To increase the confidence in the data obtained, results were set at the 1% false discovery rate (Elias & Gygi, 2007).

Normalized spectral index quantitation (SINQ) (Trudgian *et al*, 2011) was used to determine the fold changes in phosphorylation levels of proteins between HIV-1 and mock-infected DCs (Fig 1C). A total of 124 proteins were detected as being exclusively phosphorylated in DCs by HIV-1 treatment (Appendix Table S1), 134 were present in both conditions, and 48 proteins were exclusively phosphorylated in mock-infected DCs alone (Appendix Table S2). Based

on SINQ analysis, 68 proteins had > 1.25-fold higher phosphorylation levels upon HIV-1-infection (Appendix Table S3). In comparison, 66 proteins displayed > 0.75-fold less phosphorylation in the HIV-1-infected fractions, indicating that they were dephosphorylated in the presence of HIV-1 (Appendix Table S4). Thus, we identified 306 candidate proteins of a signalosome that are differentially phosphorylated upon HIV-1 infection of DCs.

A total of 167 proteins found to be differentially phosphorylated upon HIV-1 entry to DCs are known HIV-1 interacting proteins or involved in the HIV-1 life cycle based on the HIV-1 human interaction database validating our approach (Appendix Table S5). As a first step to identifying the function of the DC signalosome, we subjected the 306 proteins to pathway analysis using ingenuity pathway analysis (IPA) for molecular and cellular function (Fig 1D). The top canonical pathways in IPA were protein ubiquitination, amyloid processing, antigen presentation, glycolysis/gluconeogenesis, and insulin-like growth factor (IGF)-1 signaling (Fig 1E and Appendix Table S6).

These findings reveal that HIV-1 triggers a complex signaling event within minutes of entry to DCs that involves proteins associated with diverse cellular processes. Altogether, this indicates that active interaction with the host machinery is employed by HIV-1 likely to evade cell intrinsic innate detection by DCs and facilitate *trans*-infection.

## Discovery of proteins involved in *trans*-infection

We next investigated the role of proteins in the HIV-1 DC signalosome in HIV-1 *trans*-infection to CD4$^+$ T cells by conducting a secondary siRNA screen. Using a combination of bioinformatics and literature analysis based on the proteins detected as differentially phosphorylated in our primary screen, 125 genes with roles in either trafficking, ubiquitination, signaling, autophagy, and neural synapse formation were selected for siRNA targeting. RNA interference (RNAi) was performed to study the loss of function of these genes in HIV-1 *trans*-infection from DCs to CD4$^+$ T cells (Fig 2A and Materials and Methods). Immature DCs were transfected with siRNAs targeting the respective gene of interest (Fig 2B) for 48 h prior to HIV-1 infection for 2 h and subsequent co-culture with autologous CD4$^+$ T cells. The level of *trans*-infection was then assessed using FACS analysis of p24-positive CD4$^+$ cells (Fig 2C and D). The knockdown of four genes reduced HIV-1 *trans*-infection by more than 50% compared to controls. A further 51 genes when knocked down reduced HIV-1 *trans*-infection to between 50 and 75% of control levels (Appendix Table S7). Thus, HIV-1 entry into DCs triggers a signaling response that regulates *trans*-infection of the virus to CD4$^+$ T cells, promoting disease transmission. We performed IPA on the 55 genes affecting HIV-1 *trans*-infection and found that 26 are involved in trafficking, specifically in pathways such as integrin signaling, endocytosis, and focal adhesion kinase (FAK) signaling (Fig 2E).

## Snapin is required for *trans*-infection and influences HIV-1 localization in DCs

Of the proteins found to reduce levels of HIV-1 *trans*-infection when they were knocked down, three belonged to the BLOC-1 (biogenesis of lysosome-related organelles complex 1) complex.

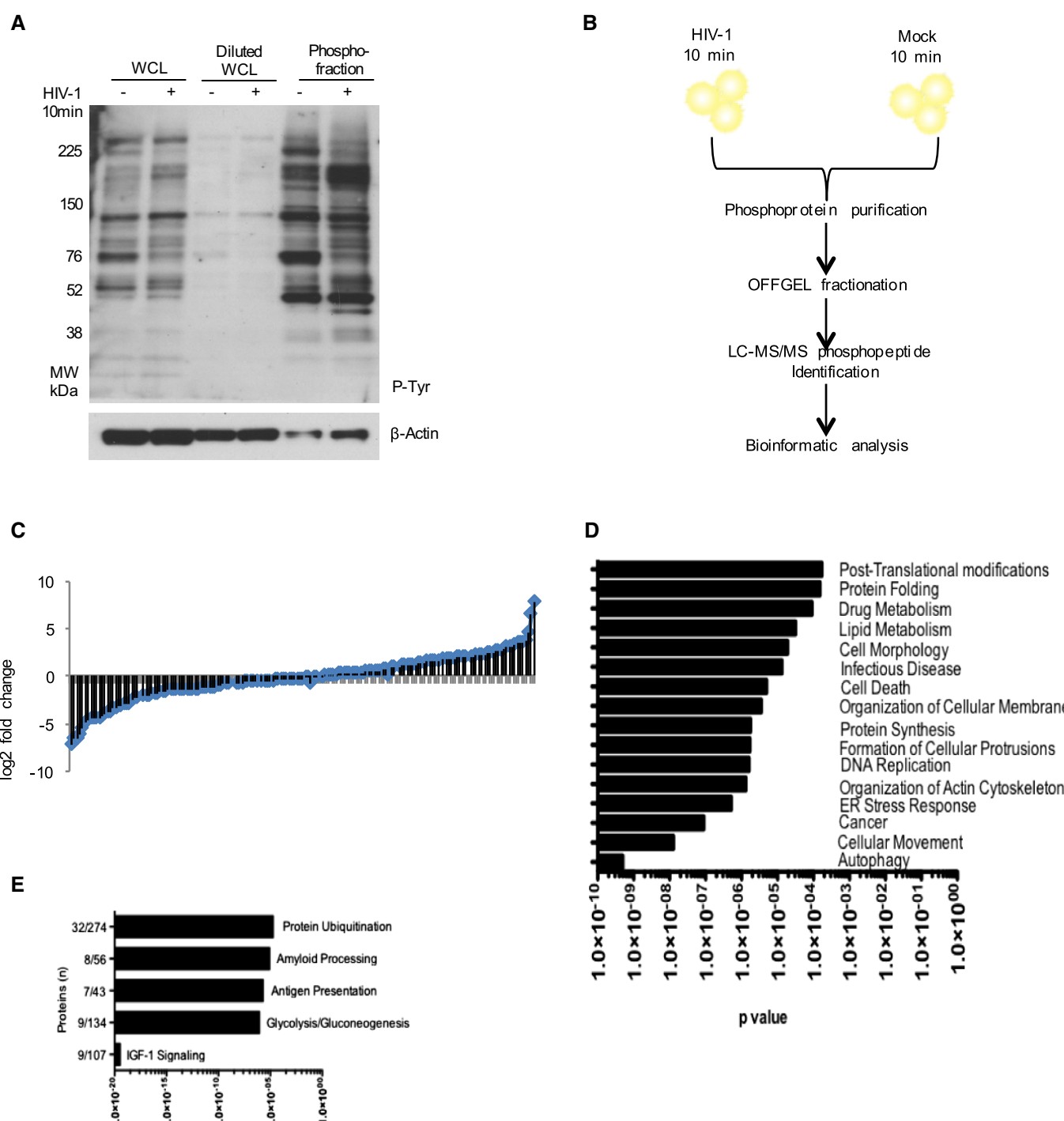

**Figure 1.  HIV-1 induces a signalosome on entry to DCs.**

A   Immunoblot using anti-phosphotyrosine antibody of DC whole-cell lysates (WCL) in the presence and absence of HIV-1 for 10 min (lanes 1 and 2), the same WCL diluted 1:10 (lanes 3 and 4), and phosphoenriched fractions derived from the same WCL (lanes 5 and 6).

B   Schematic of the phosphoproteomic approach used. DCs were exposed to HIV-1 or mock infected for 10 min, lysed and enriched for phosphoproteins. Nine biological replicates were collected, pooled, and quantified by Bradford dye assay. Equalized amounts of phosphoenriched lysates were separated by OFFGEL fractionation and prepared for LC-MS/MS analysis before bioinformatics analysis by the Central Proteomics Facilities Pipeline based in Oxford.

C   Plot showing distribution of fold changes in protein abundance between HIV-1 and mock-infected samples as determined by SINQ. The $x$-axis (blue rhombi) represents proteins identified and the $y$-axis (black bars) the associated log2 fold change of the abundance of the protein infected as opposed to mock-infected fractions.

D, E   Ingenuity pathway analysis (IPA) of the HIV-1 phosphoproteome in DCs showing molecular and cellular function (D) and top five canonical pathways (E).

Source data are available online for this figure.

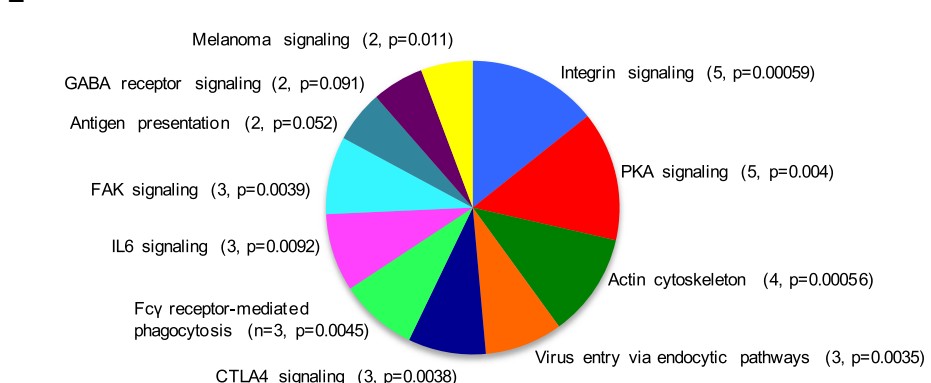

**Figure 2.**

**Figure 2. Secondary siRNA screen utilizing a FACS-based assay of *trans*-infection as readout.**

A   Workflow for the secondary RNAi screen is shown. $5 \times 10^4$ DCs were transfected in triplicates with SMARTpool siRNAs and incubated in 96-well plates for 48 h. Cells were then washed and exposed to HIV-1 NL4-3 BaL for 2 h prior to co-culture with autologous CD4$^+$ PHA blasts. Three days later cells were stained with surface markers and intracellular p24 and analyzed by flow cytometry for intracellular p24 gag in CD4$^+$ T cells.

B   Knockdown efficiency of selected genes. DCs were electroporated with non-silencing and specific siRNAs using the Neon transfection system and harvested 48 h later for immunoblot of proteins as indicated.

C   Representative FACS plots of siRNAs in the screen showing reduction (siRNA 1) or no effect (siRNA 10) on HIV-1 *trans*-infection.

D   Dot-plot of RNAi screen for identification of genes affecting HIV-1 *trans*-infection. Each dot represents a single gene. Values are mean of three independent experiments from different donors. The horizontal line indicates the population mean.

E   Ingenuity pathway analysis classifying host factors affecting HIV-1 *trans*-infection by molecular function. Only categories containing 2 or more factors were charted. Number indicates number of genes identified per category.

Phosphoproteomic analysis had also identified three components of the BLOC1 complex to be differentially phosphorylated within 10 min of HIV-1 infection. These were BLOC1 subunit 1 (BLOC1S1), which was dephosphorylated, and cappuccino homolog (CNO) and pallidin homolog (PLDN), which were phosphorylated. The BLOC-1 complex belongs to the endosomal protein sorting machinery and was first identified through its association with Hermansky–Pudlak syndrome (HPS), a group of heterogeneous autosomal-recessive disorders caused by mutations in genes affecting the biogenesis of lysosomal-related organelles (LROs). BLOC-1 is an octameric complex containing two stable sub-complexes, pallidin-Cappuccino-BLOS1 and dysbindin-Snapin-BLOS2 (Starcevic & Dell'Angelica, 2004). We examined the effect of BLOC-1 subunits on HIV-1 *trans*-infection and found that knockdown of four members, specifically muted, dysbindin, BLOS3, and the dynein motor protein Snapin had a significant effect on HIV-1 *trans*-infection (Fig 3A). Snapin knockdown had no significant effect on HIV-1 internalization as assessed by FACS analysis of DCs transfected with Ctrl or Snapin siRNAs and exposed to ATTO 488 labeled HIV-1 for 2 h (Appendix Fig S2). Snapin coordinates retrograde transport and late endosomal trafficking by directing vesicle transport and fusion (Cai *et al*, 2010). It is expressed in most cell types and is important for the release of neurotransmitters in the nervous system (Ilardi *et al*, 1999; Buxton *et al*, 2003). In our assay, knockdown of Snapin had the most dramatic effect, reducing *trans*-infection by 50%, and was therefore investigated further.

We first analyzed whether HIV-1 localizes with Snapin by infecting DCs with a FLAG-Snapin lentivirus followed by infection with HIV-1 for 10 or 30 min. We detected co-localization of HIV-1 with Snapin-FLAG already with 10 min of infection in DCs (Fig 3B). In neurons, Snapin is essential for endosomal vesicle maturation by linking late endosomal transport and the retrograde transport motor dynein. Snapin knockout neurons accumulate immature lysosomes (Cai *et al*, 2010). Two residues in Snapin, V92 and L99, have been implicated in dynein binding. We therefore explored whether this dynein binding region was required for the effect on HIV-1 *trans*-infection. DCs were infected with a lentiviral expression vector encoding wild-type (WT) Snapin-FLAG, and either of two lentiviral vectors encoding mutations in the dynein binding region V92K and L99K (Cai *et al*, 2010) and Appendix Fig S3, and analyzed for HIV-1 *trans*-infection capacity. A significant reduction in *trans*-infection was observed following overexpression of the dynein binding defective mutants V92K and L99K in DCs (Fig 3C). Snapin has been implicated in endosomal vesicle maturation and is required for synaptic vesicle release and maintenance of the recycling vesicle pool. Therefore, we explored the role of Snapin in the localization of HIV-1 within specific vesicle subtypes in DCs. We infected DCs with a DsRed Snapin shRNA and a DsRed control (Ctrl) shRNA and verified knockdown at both the mRNA and protein levels (Fig 3D and E). Confocal microscopy was used to examine the localization of the HIV-1 p18 gag protein within early endosomal (EEA-1) or late endosomal (Rab7) compartments. Knockdown of Snapin resulted in accumulation of HIV-1 specifically within the early EEA-1-positive endosomal compartment (Fig 3F). Conversely reduced levels of HIV-1 could be detected in the late endosomal (Rab7 positive) compartment (Fig 3G). In addition following Snapin knockdown, there appeared

**Figure 3. Snapin is required for HIV-1 *trans*-infection.**

A   DCs were transfected with Ctrl siRNAs or siRNAs targeting subunits of the BLOC1 complex (Blos1, Blos2, Blos3, Cappacino (CNO), Dysbindin (DTNBP1), Muted, Palladin (PLDN), or Snapin) prior to undertaking the FACS-based *trans*-infection assay described in Fig 2.

B   DCs were transfected with Flag-tagged Snapin for 24 h prior to exposure to HIV-1 at 10 and 30 min. Left panel shows confocal images of DCs stained with anti-p24 (green), anti-Flag (red), and Topra nuclear stain (blue). Right panel shows plot of correlation coefficient of HIV-1 co-localized with Snapin over time.

C   DCs transfected with Ctrl, Flag-tagged wild-type (WT), mutant (L99K), or (V92K) Snapin for 24 h prior to exposure to HIV-1 and co-culture with CD4$^+$ T cells. p24 levels detected in DC CD4$^+$ T cell co-cultures are shown.

D   DCs transfected with Ctrl or Snapin siRNAs for 24 h and qPCR for Snapin shown.

E   DCs transfected with Ctrl or Snapin siRNAs for 24 h and immunoblot for Snapin (Actin was used as a loading control).

F   DCs transfected as in (D) and stained with anti-p24 (green), anti-EEA1 (red), and Topra nuclear stain (blue). Representative confocal images are shown. Graph shows co-localization between HIV-1 and EEA-1 in Ctrl or Snapin knockdown cells.

G   DCs transfected as in (D) and stained with anti-p24 (green), anti-Rab7 (red), and Topra nuclear stain (blue). Graph shows co-localization between HIV-1 and Rab7 in Ctrl or Snapin knockdown cells.

H   DCs transfected with Flag-tagged wild-type (WT) or mutant (L99K) or (V92K) Snapin lentivectors for 24 h and stained with anti-Flag (pink), anti-p24 (green), anti-EEA1 (red), and Topra nuclear stain (blue). Graph shows co-localization between HIV-1 and EEA-1 in WT, V92K, or L99K transfected cells.

Data information: Data are representative of three independent experiments from different donors. Results are presented as mean ± SEM. Co-localization analysis of the acquired images was performed using Fiji software with calculation of Pearson's and Li's coefficient. Statistical analysis was performed using unpaired Student's *t*-test. *$P < 0.05$, **$P < 0.01$, ***$P < 0.001$, ****$P < 0.0001$. Scale bars represent 6 μm.

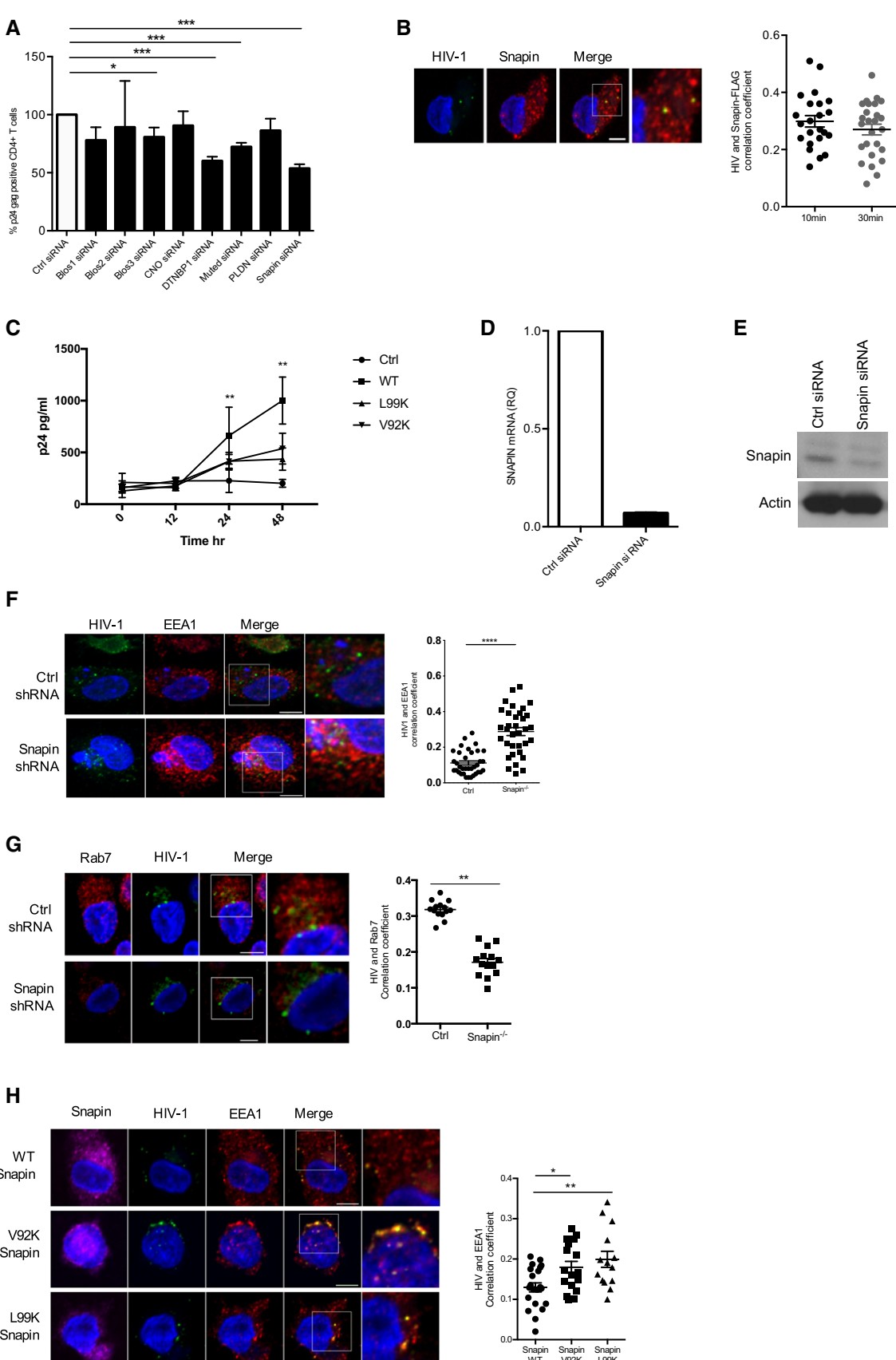

**Figure 3.**

to be a different morphological appearance of EEA1 and Rab7 vesicles within DCs; however, the overall intensity of EEA1 or Rab7 staining did not change (Appendix Fig S4). Further, we infected DCs with a Snapin-FLAG wild-type, V92K- or L99K-dynein binding mutant lentiviral expression vector as before. Confocal experiments showed that the retention of HIV-1 in the early EEA-1-positive compartment is mediated via the dynein binding region of Snapin (Fig 3H). Thus, Snapin functions in HIV-1 trafficking from early-to-late endosomes in DCs, and *trans*-infection to T cells, which may be mediated by dynein binding.

## Snapin depletion leads to enhanced innate responses to HIV-1 in DCs

Our results thus far have identified components of the BLOC-1 complex, which functions in endosomal sorting, as being differentially phosphorylated upon HIV-1 infection of DCs, and specifically Snapin as a key regulator of HIV-1 intracellular trafficking and *trans*-infection. Given HIV-1 is able to evade detection in the endosome, we next examined the role of Snapin in HIV-1-sensing and TLR8 signaling. We therefore examined the ability of Snapin to

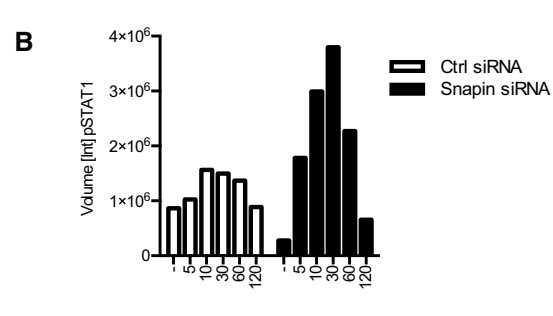

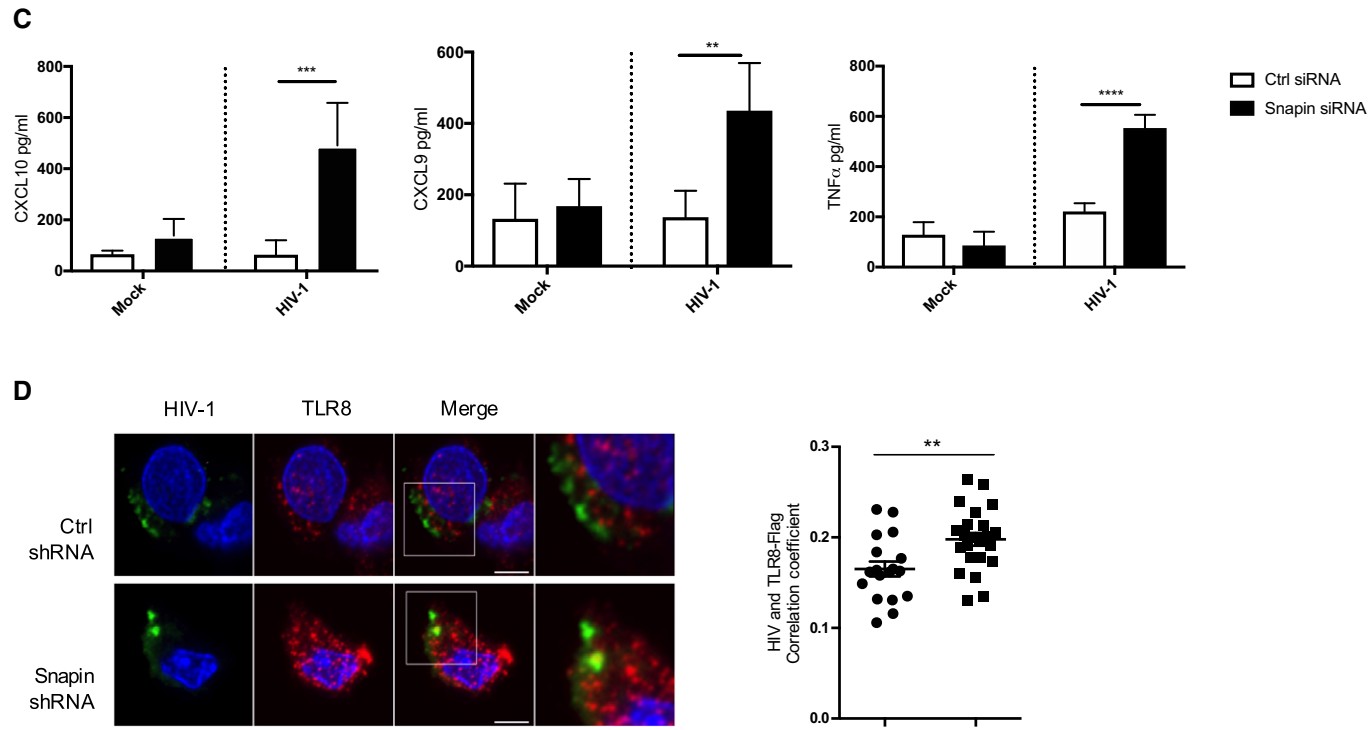

**Figure 4.  Snapin influences HIV-1 localization with TLR8 and signaling.**

A   DCs transfected with Ctrl or Snapin siRNAs for 24 h prior to exposure to HIV-1 over time, lysed and immunoblot undertaken using anti-pStat1 or anti-actin antibody.
B   For quantification, images were acquired with the ChemiDoc MP system and analysis was performed using Image Lab 4.0 software.
C   ELISAs of CXCL10, CXCL9 and TNFα levels detected following HIV-1 exposure to Ctrl or Snapin knockdown DCs.
D   MUTZ3 cells were transduced with a TLR8-FLAG lentivirus followed by Ctrl or Snapin shRNA and analyzed by confocal microscopy for co-localization of HIV-1 with TLR8-FLAG 1 h post-infection (cells stained with anti-p24 (green) and anti-Flag (red)). Right panel shows degree of co-localization.

Data information: Data are representative of three independent experiments from different donors. Results are presented as mean ± SEM. Co-localization analysis of the acquired images was performed using Fiji software with calculation of Pearson's and Li's coefficient. Statistical analysis was performed using unpaired Student's *t*-test. **$P < 0.01$, ***$P < 0.001$, ****$P < 0.0001$. Scale bar represents 6 μm.

Source data are available online for this figure.

                    

influence activation of Stat1 in DCs upon HIV-1 exposure by transfecting immature DCs with Ctrl or Snapin siRNAs prior to lysis and immunoblotting for phospho-Stat1. Snapin knockdown enhanced phospho-Stat1 levels as compared to control siRNA-transfected cells (Fig 4A and B). In addition, Snapin knockdown led to enhanced release of CXCL10, CXCL9, and TNFα from HIV-1-exposed DCs (Fig 4C). This was not due to increased viral replication, as qPCR of early expressed HIV-1 transcripts revealed no difference between Ctrl and Snapin knockdown cells (Appendix Fig S5). Furthermore depletion of TLR8 in Snapin knockdown cells impaired the enhanced release of TNFα (Appendix Fig S6) suggesting that Snapin inhibits immune signaling through TLR8 in HIV-1-infected DCs. To investigate how, we analyzed HIV-1 and TLR8 localization in the presence and absence of Snapin. We used MUTZ-3 cells, a human dendritic cell line, expressing TLR8-V5 and transduced with dsRed Ctrl or Snapin shRNA lentiviral vectors. Co-localization of HIV-1 with TLR8 was significantly increased in the absence of Snapin (Fig 4D). These data suggest that Snapin inhibits TLR8-mediated Stat1 signaling to HIV-1 in DCs by promoting HIV-1 intracellular trafficking from early-to-late endosomes.

### Snapin downregulates TLR8 signaling in DCs

To explore whether this new function for Snapin is specific to HIV-1 or whether it exerts a more generic effect on TLR8 signaling in DCs, we tested whether Snapin can regulate TLR8 signaling in response to stimulation by other TLR8 ligands. ssRNA40 is a guanosine- and uridine-rich single-stranded oligonucleotide derived from HIV-1 and an established ligand for TLR8 in DCs (Heil *et al*, 2004). We detected increased IL-6 and IFNβ1 mRNA levels and increased IL-6 and TNFα protein levels following ssRNA40 stimulation (Fig 5A and B). Importantly, these increases were significantly enhanced upon Snapin knockdown. R848 is an imidazoquinoline compound with potent anti-viral activity and an unrelated TLR8 agonist. R848 and ssRNA40 induce differing signaling cascades downstream of TLR8, with R848 preferentially activating a p38 cascade and ssRNA40 strongly activating ERK (Colak *et al*, 2014). We examined these signaling cascades in the presence and absence of Snapin knockdown and found that loss of Snapin enhanced phosphorylation of p38 and STAT1 in R848 stimulated DCs, and enhanced phosphorylation of p38 and ERK in ssRNA40 stimulated DCs (Fig 5C and D). To determine whether Snapin regulated PRR signaling, more broadly we examined pro-inflammatory cytokine responses following stimulation of surface localized TLR2 in Ctrl or Snapin knockdown DCs. No significant changes in the level of

response to TLR2 stimulation in the absence of Snapin were observed suggesting the effect may be compartment specific. We also tested the effect of Snapin knockdown on responses to the paramyxovirus Sendai virus that is sensed by TLR7/8 in myeloid cells (Melchjorsen *et al*, 2005). Here, we observed increased induction of *TNFA* and *ifnb1* mRNA and TNFα protein in Snapin knockdown cells following infection with Sendai virus (Fig 5E). Thus, Snapin downregulates TLR8 signaling to attenuate pro-inflammatory responses in myeloid cells.

### Snapin influences endocytosed bead localization and endocytic vesicle maturation in DCs

To explore how Snapin exerts this general effect on TLR8 signaling, we investigated its role in endocytic vesicle maturation in DCs in more detail. We first tested whether Snapin affected the intracellular localization of endocytosed beads in DCs in a similar manner to its effect on HIV-1 virions. We exposed DCs to labeled beads and measured their localization with EEA-1 early endosomes at 1-hr post-exposure in Ctrl shRNA or Snapin shRNA-transfected DCs. Similar to HIV-1, Snapin knockdown enhanced localization of internalized beads with EEA-1 (Fig 6A). We then tested whether Snapin affects the rate of vesicle maturation in human DCs. pHrodo® Red dextran has a pH-sensitive fluorescence emission that increases in intensity with increasing acidity. This increase is particularly marked in the pH range 5–8, commonly seen as endocytic vesicles are acidified. We exposed Ctrl or Snapin shRNA-transduced DCs to pHrodo® Red dextran and undertook FACS analysis of the red-fluorescent signal at time points over 1 h. The fluorescent signal was reduced in Snapin knockout DCs indicating decreased acidification of endosomes in keeping with the delay in trafficking to late endosomes observed by confocal microscopy (Fig 6B). Thus, Snapin directs endocytic vesicle maturation more generally in DCs independent of TLR8 triggering or HIV-1 exposure.

## Discussion

Our proteomic screen revealed that HIV-1 induces a robust signaling event on entry to human DCs, in contrast to its lack of ability to induce vociferous cell intrinsic innate immune responses in these cells. Combining this approach with a secondary focused siRNA screen enabled the identification of numerous new host cell mediators of HIV-1 *trans*-infection in DCs, which will act as a valuable resource for future studies.

---

**Figure 5.  Enhanced signaling through TLR8 in Snapin knockdown cells.**

A   DCs were transfected with Snapin or Ctrl siRNAs and activated with ssRNA40. mRNA was harvested 6 h post-activation for IL-6 and IFNβ1 qPCR. Results were relative to a GAPDH and lyovec control.

B   DCs treated as in (A) and supernatant analyzed using IL-6 or TNFα ELISA. Data are representative of 4–6 independent experiments.

C   DCs transfected with Ctrl or Snapin siRNAs and stimulated with R848 over time. Cells were lysed and immunoblotted for phospho-p38 (pp38), phospho-Stat1 (p-STAT1), STAT1, phospho-IκB-α, and LAMP1. Immunoblot quantification is shown in the right panel.

D   DCs transfected with Ctrl or Snapin siRNA and activated with ssRNA40 over time. Immunoblot using anti-phospho-p38 (upper panel), phospho-Erk (middle panel), and β-tubulin (lower panel) is shown. Immunoblot quantification is shown in the right panel.

E   mRNA of *TNFA* and *ifnb1* (left and middle panel) or TNFα protein level in DCs transfected with Ctrl or Snapin siRNAs and left uninfected or infected with Sendai virus.

Data information: Data are representative of three independent experiments from different donors. Results are presented as mean ± SEM. *P < 0.05, ***P < 0.001, ****P < 0.0001. Statistical analysis was performed using unpaired *t*-test.
Source data are available online for this figure.

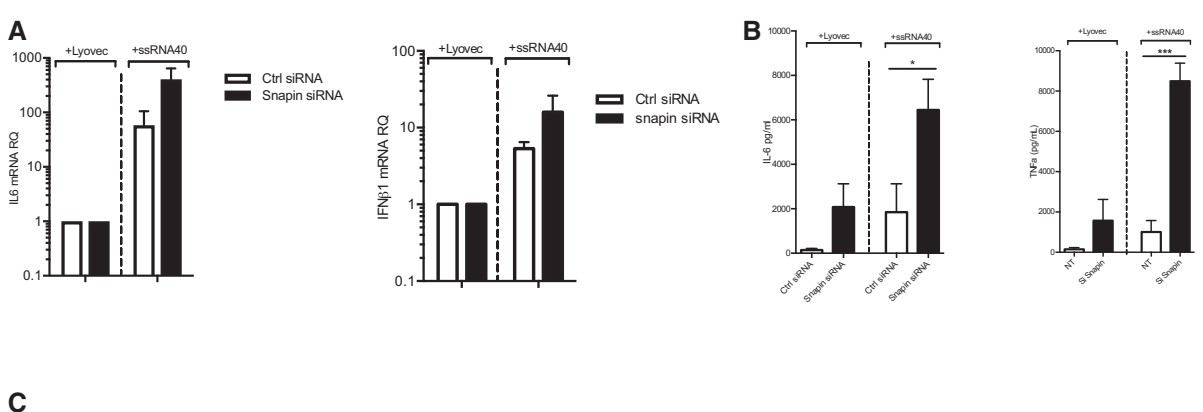

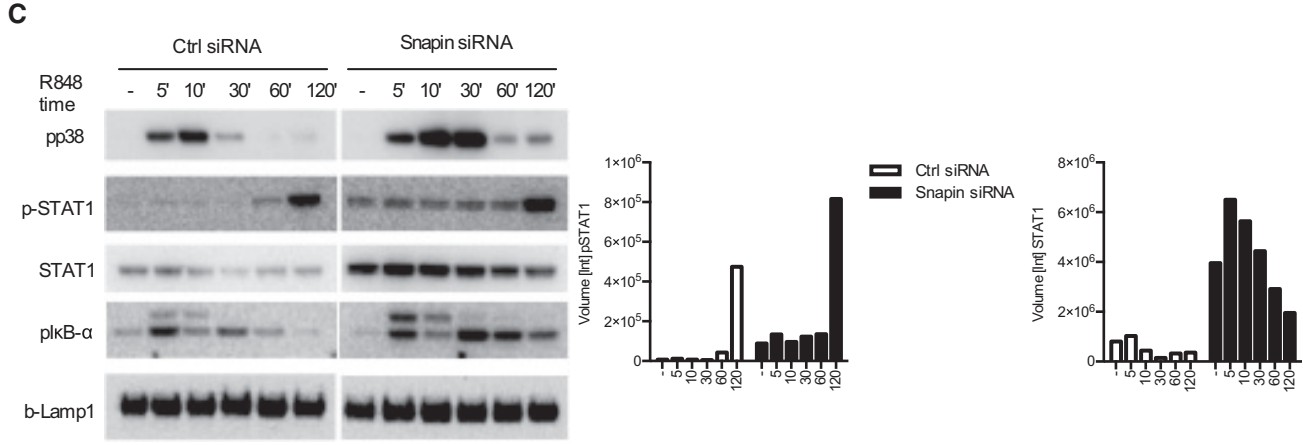

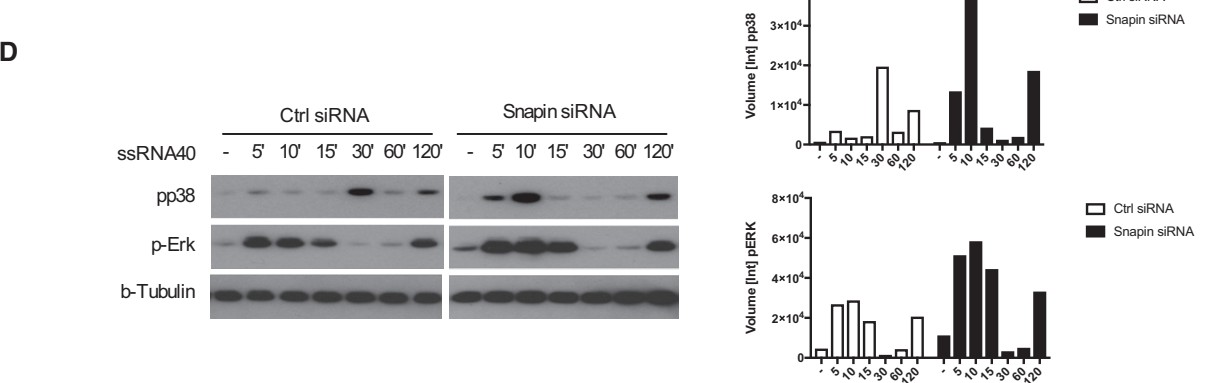

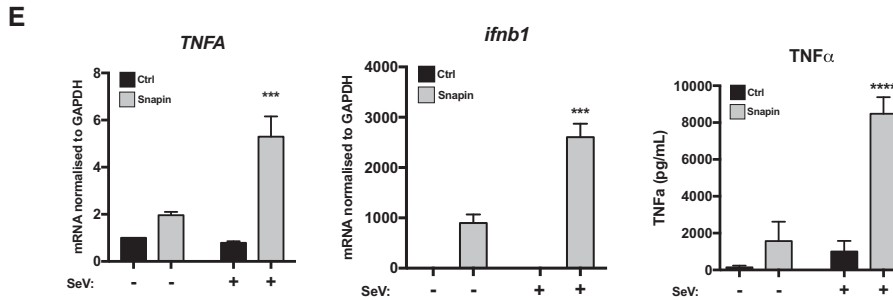

**Figure 5.**

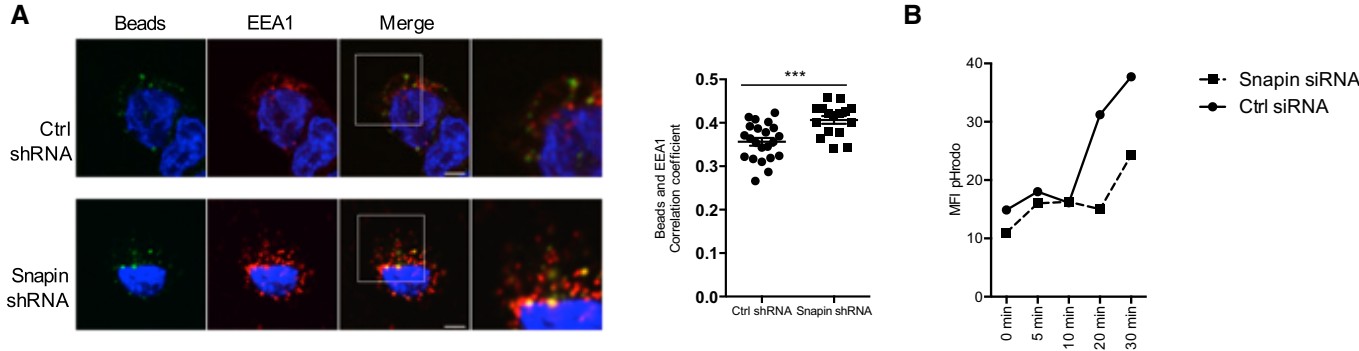

**Figure 6.  Snapin is required for maturation of early endosomes.**

A   DCs transfected with Ctrl or Snapin siRNAs for 24 h prior to incubation with beads. Left panel shows confocal images (beads (green), anti-EEA1 (red), nuclear stain (blue)), and right panel shows degree of co-localization of beads with EEA1.

B   DCs transfected with Ctrl or Snapin siRNAs for 24 h prior to incubation with pH-sensitive pHrodo™ Green dextran (10,000 MW, P35368, 509/533 nm) for different time points prior to FACS analysis.

Data information: Data are representative of three independent experiments from different donors. Results are presented as mean ± SEM. Co-localization analysis of the acquired images was performed using Fiji software with calculation of Pearson's and Li's coefficient. Statistical analysis was performed using unpaired Student's *t*-test. ***$P < 0.001$. Scale bar represents 6 μm.

One of the proteins most profoundly affecting *trans*-infection was Snapin, a component of the BLOC-1 complex, which is involved in endosomal protein sorting. Early endosomes represent the first sorting station of newly internalized material, which includes pathogens, in all eukaryotic cells. Depending on its final destination, the material is either packed into budding vesicles or remains within early endosomes until they mature into late endosomes (marked by Rab7) and then fuse with lysosomes for degradation of their content. We have identified a general role for Snapin in TLR8-mediated immune signaling. Snapin functions by promoting maturation of EEA1 early endosomes.

In this role, Snapin inhibits HIV-1 detection by TLR8 in endosomes and subsequently promotes *trans*-infection and viral spread through the host. Although this study focused on defining a role for Snapin in endosomal maturation and TLR signaling, it is also possible Snapin may be involved in secretion of HIV-1 from DCs across the viral synapse akin to its previously described role at neurological synapses (Ilardi *et al*, 1999; Buxton *et al*, 2003). While we observed enhanced co-localization of HIV-1 with EEA1-positive vesicles following depletion of Snapin, the morphological appearance of EEA1 appeared to subtly change between Ctrl and Snapin knockdown cells. It is possible as part of the BLOC1 complex Snapin knockdown leads to a failure of fragmentation or enhanced fusion of early endosomes in addition to its role in TLR8 sensing.

TLRs are attractive therapeutic targets for the modulation of immune responses and hold promise for the treatment of infection and inflammation. Agonists of TLR7 and TLR8 including imidazoquinolines are currently being used or tested in clinical trials as anti-viral and anti-cancer drugs (Iyer, 2015; Peris & Fargnoli, 2015). Shedding light on the mechanism of signaling of imidazoquinolines is important to maximize their potential as therapeutic agents. There is mounting evidence that both subcellular localization and endosomal pH can influence the ability of endosomal TLRs to signal. For example, the TLR9 response to exogenous CpG DNA is dependent on specific endosomal localization and acidification (Guiducci *et al*, 2006; Yao *et al*, 2009; Okuya *et al*, 2010; Hazeki *et al*, 2013;

Duhamel *et al*, 2016). Sasai *et al* (2010) showed TLR9 signaling leading to activation of type I IFN but not pro-inflammatory cytokine genes require TLR9 trafficking from endosomes to a specialized lysosome-related organelle dependent on AP-3. AP-3 has additionally been shown to enhance phagosomal TLR4 signaling in DCs and antigen presentation to CD4[+] T cells. Peptide: MHC-II export to the cell surface was impeded in AP-3-deficient mice correlating with reduced TLR4 recruitment and pro-inflammatory signaling from phagosomes (Mantegazza *et al*, 2012). In our work, we show for the first time that TLR8 signaling also depends on correct subcellular localization and demonstrate that Snapin is essential for endolysosomal compartment acidification in DCs. Taken together, this supports a model where Snapin marks an immune-regulatory compartment in DCs that regulates innate immune sensing by TLR8 via its control of endocytic vesicle maturation.

Our data suggest HIV-1 preferentially localizes with Snapin following entry to DCs as a mechanism to evade immune detection. How this occurs is currently unclear. It is possible HIV-1, by binding to specific tolerogenic entry receptors such as DC-SIGN, activates PKA to phosphorylate Snapin and secure Snapin binding to the SNARE complex. This would result in preferential internalization of HIV-1 in Snapin-positive vesicles destined for maturation or recycling.

Mutations in genes encoding proteins of the BLOC-1 complex are defective in inbred mouse strains that serve as models of HPS, a rare autosomal-recessive disorder characterized by hypopigmentation and platelet storage pool deficiency and also associated with immunodeficiencies (Dotta *et al*, 2013). The HPS protein AP-3 has already been shown to affect the late stages of the HIV-1 replication cycle, with the matrix region of Gag interacting directly with the delta subunit of the AP-3 complex to promote Gag trafficking in human embryonic kidney 293T cells (Dong *et al*, 2005). Our work defines a new role for HPS proteins in innate immunity. It will be interesting to explore whether defects in endosomal TLR signaling are observed in subsets of HPS patients with Snapin or associated protein mutations, which may contribute to inflammatory phenotypes such as colitis observed in this disease.

   

## Materials and Methods

### Cell culture, stimulation, and HIV-1 infection

Human THP-1 cells (myelomonocytic) from ATCC were cultured in R10 (RPMI supplemented with 10% (v/v) heat-inactivated FCS, penicillin/streptomycin, L-glutamine, 25 mM HEPES, nonessential amino acids, 1 mM sodium pyruvate, 50 μM β-mercaptoethanol, and NaHCO3 (Sigma)), and HEK293T cells were cultured in D10 (DMEM high glucose supplemented with 10% (v/v) heat-inactivated FCS, penicillin/streptomycin, and L-glutamine). Cells were grown at 37°C in humidified air with 5% $CO_2$. If not indicated otherwise, monocytic THP-1 cells were differentiated into a macrophage-like cell line by culturing them in R10 supplemented with 10–15 nM PMA for 24–72 h. Buffy coats were obtained from the National Blood Centre (Bristol, UK) following local ethical guidelines granted by Milton Keynes Research Ethics Committee (Ref 07/H0603/43). $CD14^+$ monocytes were positively selected (Miltenyi Biotech) from peripheral blood mononuclear cells (PBMCs) and cultured with 40 ng/ml recombinant IL-4 and GM-CSF (Peprotech) for 5 days. Purity immature dendritic cells were harvested on day 4. HIV-1 NL4-3 BaL virus stocks were titrated by p24 ELISA (Immunodiagnostics and ABL). $TCID_{50}$ of viral stock was measured by limiting dilution infectivity assay on TZM-bl cells (human cervical cancer cells from AIDS Research and Reference Reagent Program). MDDCs were pulsed with titrated HIV-1 at an MOI of 0.1–1. Excessive virus was washed off with RPMI (Sigma) supplemented with RPMI supplemented with 10% (v/v) FCS (Sigma), 2 mM L-glutamine, 1 U/ml penicillin, and 50 μg/ml streptomycin after 120 min. $CD4^+$ T cells were negatively isolated by negative immunomagnetic bead selection (Miltenyi Biotech). HIV-1 infected MDDCs were co-cultured with PHA-blasted $CD4^+$ T cells at a ratio of 1:3 (DC:CD4) at $2 \times 10^6$ cells/ml. MUTZ-3 cells, a human dendritic cell line, were kindly provided by Professor T.D. Gruijl and cultured in conditioned medium from 5,637 cells. ATTO-488-maleimide (ATTO-Tec GmbH, Sigma-Aldrich) was used for labeling of HIV-1 virus. Dye was reconstituted in DMSO according to manufacturer's instructions and then added dropwise with intermittent vortexing to purified, concentrated HIV-1. The mixture was incubated light protected at RT for 2 h before being diluted with PBS, underlay with 20% sucrose, and purified by centrifugation.

### RNAi screen

MDDCs were prepared and resuspended at 25 mill/ml in transfection solution for Neon® transfection using 10 μl tips (Thermo Fisher). ON-TARGETplus SMARTpool siRNAs (20 μM stock) were obtained from Dharmacon and used according to the manufacturer's recommendations. Electroporation settings were as follows: pulse voltage 1,475 V, pulse width 20 ms, pulse number 2. Transfections were done in three technical replicates per gene. Following electroporation, cells were cultured for 48 h to allow knockdown of gene expression before infection with HIV-1 for 2 h followed by two washes. DCs were co-cultured with autologous PHA-blasted $CD4^+$ T cells for a further 72 h. For flow cytometry analysis, cells were stained with anti-LIVE/DEAD stain, anti-CD3, and intracellular HIV-1 p24 after treatment with BD cytofix/cytoperm buffer. Controls used were HIV-1 uninfected and non-targeting siRNA (Dharmacon). Data were analyzed using Flowjo software.

### Confocal microscopy

Transfected cells or THP-1 cells were seeded onto eight-well poly-D-lysine Tissue Culture Slides (BD Biosciences) at $2 \times 10^5$ cells/well before infection with HIV-1 at indicated time points. Samples were fixed with 4% paraformaldehyde (Sigma) for 15 min at room temperature and permeabilized twice for 1 min with 0.5% Triton X-100. Blocking buffer (PBS containing 5% human serum, PAA; 5% FCS, 1% BSA, 5% goat serum, and 1:20 diluted HuFcR Binding Inhibitor (eBioscience)) was added for 1–2 h before samples were treated with primary antibodies in blocking buffer for 30 min. Samples were washed three times with PBS and stained with secondary antibodies in blocking buffer for 30 min, washed again as before, and then mounted using fluoromount with or without DAPI (Southern Biotech). All images were taken to avoid saturation using a confocal microscope (Zeiss 780; Carl Zeiss) equipped with a chromatically corrected Apochromat 63× lens. This lens is designed to minimize chromatic aberration, which makes it ideal for co-localization studies. Analysis was done on raw image data. For quantitative co-localization analysis, Li's coefficient was calculated using the application Coloc2 in Fiji (Schneider & Eliceiri, 2012) (http://imagej.nih.gov). Such analysis allows characterizing the degree of overlap between two channels in the multidimensional microscopy image recorded at different emission wavelengths.

### Flow cytometry analysis to measure endocytosis of pHrodo™ dextran

Experimental procedure was performed according to manufacturer's instructions. In brief, cells were washed and growth medium was replaced with X-Vivo (Lonza). pHrodo™ dextran was added to the cells at a final concentration of 20–100 μg/ml, and cells were incubated at 37°C for the indicated time points. Cells were washed with pre-warmed, dye-free medium at pH 7.4. Cells were returned to dye-free medium at pH 7.4, and flow cytometric analysis was performed.

### Phosphoprotein purification

Immature MDDCs were exposed to HIV-1 NL4-3 BaL or mock for 10 min and washed twice before lysis and phosphoprotein enrichment according to the to the manufacturer's instructions (Qiagen). Lysates were desalted using ZEBA Spin desalting columns (Thermo Fisher), concentrated, and fractionated iso-electric focusing (Agilent 3100 OFFGEL High Res kit, pH 3–10) resulting in twelve fractions per condition. Before mass-spectrometry fractions were subjected to methanol–chloroform precipitation and in-solution tryptic digestion and Sep-Pak purification (Waters).

### Mass-spectrometry analysis

LC-MS/MS was performed out in collaboration with Dr Benedikt Kessler (Core Proteomics Facility, University of Oxford). Mass-spectrometry data were analyzed using the Central Proteomics Facility

Pipeline (CBRG, Oxford) (Trudgian *et al*, 2010). Peptides and proteins were identified by a combination of three different search engines (Mascot, OMSSA, X!Tandem kscore). Spectra were searched with trypsin specificity and stringent filtering criteria such as cysteine carbamidomethylation and phosphorylated residues (Ser, Thr, Tyr). A false discovery rate of maximum 1% was estimated to generate high confidence data. In addition, SINQ was used to provide a semi-quantitative comparison of protein amounts between samples.

### Cell lysis and quantitative immunoblot analysis

One million cells were lysed in 50 µl lysis buffer consisting of 20 mM Tris–Cl pH 7.5, 150 mM NaCl, 1 mM EDTA pH 8.0, 1% (v/v) Triton X-100, 1 mM EGTA, 2.5 mM sodium-pyrophosphate decahydrate, and 1 mM β-glycerophosphate supplemented with complete protease inhibitors (Roche). Alternatively, lysis was carried out in 1% Nonidet P-40 lysis buffer in the presence of protease inhibitors (Roche). Proteins were separated by SDS–PAGE, transferred to PVDF membranes (Hybond-P, Amersham Biosciences), blocked for 1 h with 5% (wt/v) skim milk in PBS with 0.5% (v/v) Tween-20 and were probed for 1 h with the appropriate antibodies. Membranes were washed three times with PBS with 0.5% (v/v) Tween-20 and were incubated for 1 h with horseradish peroxidase-conjugated secondary antibody. After washing, proteins were visualized with an enhanced chemiluminescence detection reagent (Pierce). Quantification was acquired with the ChemiDoc MP system, and analysis was performed using Image Lab 4.0 software or ImageJ software.

### Real-time PCR

RNA was extracted using RNeasy Mini kit (Qiagen) and subjected to DNAse treatments (Applied Biosystems) according to the manufacturer's instructions. Complementary DNA (cDNA) was synthesized using the High Capacity RNA-to-cDNA kit (Applied Biosystems). Real-time RT–PCR was performed with pre-designed TaqMan primers (Applied Biosystems). GAPDH (glyceraldehyde 3-phosphate dehydrogenase) served as endogenous reference gene. All PCRs were set up in triplicate, and RNA expression was calculated using the $\Delta\Delta C_t$ method.

### ELISA

Transfected DCs were infected with HIV-1 or activated with PRR ligands and appropriate controls as indicated before supernatants being harvested (usually after 12 h unless otherwise indicated) and stored at −80°C until analysis. Human IL6 and TNF-α ELISA kits for cytokines were from R&D Systems and for p24 from Advances Bioscience Laboratories and used following standard protocols.

### Snapin shRNAs

The shRNA sequence targeting Snapin (UniProtKB-O95295) was designed using the siRNA at Whitehead server (Yuan *et al*, 2004) with a custom pattern to identify a suitable mRNA target, N2(GC)N (A)N6(UT)N2(ATUC)N5(A)N2, which retrieved the following sequence: 5′-GACGCGTTGTCTTGGTTAA. The shRNA expression

sequence consists of a nucleotide sense sequence identical to the target sequence described above, followed by a loop TTCAAGAGA, an antisense sequence and a stretch of five Ts. Complementary overhangs from BamHI and EcoRI were included to allow insertion into appropriate plasmid, downstream of the U6 promoter sequence. The final DNA sense and complimentary antisense sequence were ordered as custom-designed DNA oligos (Sigma-Aldrich). The two DNA oligos were mixed at a 1:1 molar ratio, heated to 95°C, and were allowed to cool slowly to 10°C. The double-stranded DNA sequence was then cloned into the pHR-U6P lentiviral plasmid. The final plasmids were sequenced to confirm DNA identity and integrity.

### Generation of lentiviruses

HEK 293T cells HEK-293T cells were transfected with equal amounts of p8.91 (Zufferey *et al*, 1997), pMD-G (Naldini *et al*, 1996), and pHR-U6P-dsRed encoding the shRNA or transgene of interest using Genejuice (Novagen Merck) as per manufacturers' recommendation. Viral supernatant was harvested 48–72 h post-transfection and concentrated by ultracentrifugation. Efficient expression of the desired proteins was analyzed by flow cytometry and/or confocal microscopy using appropriate antibodies.

### Lentiviral transduction

DCs were harvested on day 3 of differentiation and transduced with concentrated lentiviral particles expressing the shRNA or transgene of interest and Vpx-VLPs in the presence of polybrene according to the protocol described by Berger *et al* (2011). Lentiviral transduction efficiency was assessed by flow cytometry on day 4 post-infection before further functional experiments.

### Reagents and antibodies

ssRNA40, R848, and other PRR ligands were from InvivoGen, phytohemagglutinin from Rembrel. Anti-FLAG, anti-EEA1, anti-Rab7, anti-SHP1, anti-RRAGA, anti-STAT1, anti-Phospho-STAT1, anti-IkBα, anti-pp38, anti-phospho-Erk were from Cell Signaling; anti-LAMP1, anti-beta-tubulin, anti-beta-actin were from Abcam; anti-phosphotyrosine (PY20) was from Zymed, anti-HIV-1 p18 (4C9) was from NIBSC. Anti-CD4 and anti-CD3 were from BD biosciences, anti-HIV-1 core antigen (KC57) was from Beckman Coulter, Violet LIVE/DEAD Cell Stain was from Invitrogen. GABARAP was from Abnova, PSMB6 and SNX5 were from Santa Cruz, BLOC1S1 was from Proteintech, Snapin was from Synaptic Systems. Anti-rabbit Alexa Fluor 555, anti-mouse Alexa Fluor 647, and streptavidin-coated Dynabeads were from Invitrogen, pHrodo from Thermo Fisher, anti-mouse HRP and anti-goat HRP were from Sigma.

### Data availability, analysis, and statistical testing

The mass-spectrometry data from this publication have been deposited to the MassIVE database (https://massive.ucsd.edu) and assigned the identifier MSV000081396. Experimental data were analyzed using Microsoft Excel 2007 or GraphPad Prism. Biological replicates are shown as mean ± SEM, and technical replicates are

as mean $\pm$ SD. Statistical analysis was performed using Student's *t*-test and analysis of variance. **$P < 0.01$, ***$P < 0.001$, N.S. (not statistically significant) $P > 0.05$.

**Expanded View** for this article is available online.

## Acknowledgements
We thank Tanja de Gruijl, VU University Medical Center, Amsterdam, for the gift of MUTZ-3 cells. E.K was funded by a Wellcome Trust Clinical Training Fellowship; D.G. by an MRC doctoral studentship; A.L. by the Bill and Melinda Gates foundation. A.S. was funded by an NIHR Research Professorship and Wellcome New Investigator Award. We acknowledge support of the MRC to A.S.; MRC and CRUK to V.C; the support from the Wolfson Imaging Centre and Oxford NIHR Biomedical Research Centre.

## Author contributions
EK and MMH designed and performed experiments, analyzed data, and wrote the paper. DG, TP, AL, RF, DM, SB, KZ and M-YS performed experiments and analyzed data. BK, SR-J and VC provided supervision. AS conceived the study, supervised, analyzed data, and wrote the paper.

## Conflict of interest
The authors declare that they have no conflict of interest.

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
