## [Review Process File · The EMBO Journal]

Manuscript EMBO-2016-95364

Snapin promotes HIV-1 transmission from dendritic cells by dampening TLR8 signaling

Elham Khatamzas, Madeleine M Hipp, Daniel Gaughan, Tica Pichulik, Alasdair Leslie, Ricardo Fernandes, Daniele Muraro, Sarah Booth, Kieran Zausmer, Mei-Yi Sun, Benedikt Kessler, Sarah Rowland-Jones, Vincenzo Cerundolo, Alison Simmons

Corresponding author: Alison Simmons, University of Oxford

Review timeline:

Submission date:	29 July 2016
Editorial Decision:	12 September 2016
Revision received:	29 March 2017
Editorial Decision:	15 May 2017
Revision received:	20 July 2017
Editorial Decision:	26 July 2017
Revision received:	05 August 2017
Accepted:	11 August 2017

Editor: Karin Dumstrei

Transaction Report:

1st Editorial Decision

12 September 2016

Thank you for submitting your manuscript to The EMBO Journal. Your study has now been seen by three referees and their comments are provided below.

The referees appreciate the analysis, but also find that we need more insight into how Snapin is involved in HIV-1 transmission and the link to TLR8 signalling. The referees raise a number of constructive comments of how the analysis could be extended. Given the referees' comments I would like to invite you to submit a revised version of the manuscript addressing the concerns raised in full. I should add that it is EMBO Journal policy to allow only a single major round of revision, and that it is therefore important to address the raised concerns at this stage.

When preparing your letter of response to the referees' comments, please bear in mind that this will form part of the Review Process File, and will therefore be available online to the community. For more details on our Transparent Editorial Process, please visit our website: http://emboj.embopress.org/about#Transparent_Process

Thank you for the opportunity to consider your work for publication. I look forward to your revision.

REFeree REPORTS

Referee #1:

The authors have used an unbiased approach to identify signaling responses in dendritic cells upon HIV-1 binding. They show that there is a different phosphorylation profile in DCs treated with HIV-1 compared to uninfected controls. They have an unbiased method to identify which proteins are differentially phosphorylated upon HIV-1 uptake. They identify Snapin, as a protein that is somehow involved in HIV-1 transmission to CD4+ T-cells. The authors link the enhancing effect of snapin to TLR8 and hypothesize that HIV-1 uses snapin to escape TLR8 mediated antiviral immunity.

However, the main conclusions in the manuscript are not supported by the data, and some findings need to be elaborated. The authors state that Snapin downregulates TLR8 signaling in response to HIV-1. However, the data presented are not convincing as the authors fail to show TLR8 involvement in signaling upon HIV-1 entry. The role of Snapin in transmission remains unclear and more detailed experiments need to be performed to understand its role. The suggestion that Snapin is required for endosomal maturation would argue against a role in enhancing transmission as HIV-1 targeting to lysosomes will decrease virus transmission.

Major concerns

1. The authors suggest that Snapin as part of BLOC-1 complex is involved in HIV-1 transmission as silencing of Snapin decreased transmission. (Fig 3). Fig 3, only relative values are given, what is the amount of transmission (as in Fig 2, 8% transmission is low)? In Fig 3B there is hardly any virus taken up. It is difficult to assess whether this is indeed co-localization. The fact that Snapin is highly expressed in vesicles suggests that most endosomes contain Snapin, which explains the co-localisation but this does not mean that HIV-1 is specifically targeted to Snapin.
2. HIV-1 uptake into Since the authors show that snapin enables the maturation of endosomes, and that snapin knockdown reduces HIV-1 transmission, it suggests that HIV-1 benefits from being taken up in late endosomes. This is not logical, as blocking endosomal maturation would prevent HIV-1 degradation. The authors need to show that the transmission is indeed enhanced because of snapin-induced endosomal maturation. As Snapin is also involved in secretion of vesicles, this might also be an important factor in facilitating transmission. Experiments need to be performed to further investigate this.
3. What exactly is the mechanism of the enhanced transmission? The authors should elaborate on the effect of snapin silencing on infection of DCs. Show mechanism of enhanced transmission (is it cis/trans? Is uptake enhanced?) It is unclear in which step of transmission snapin is involved. On the transmission: please include controls that there is no cis-infection /transmission, for instance inhibitors.
4. Fig3C. the transmission is very difficult to assess here, controls are missing (control-transfected DCs, as well as normal DCs) and absolute values need to be given. Figure 3C Why is there no reduction with the L99K mutant? (both are mutations in the dynein binding site)
5. The colocalisation data on the confocal images are not convincing. Snapin silencing affects EEA1 and Rab7 expression, which can greatly affect co-localisation. More EEA1 expression can lead to more HIV-1 co-localisation and vice versa less Rab7 will lower the chance of co-localisation with HIV-1.
6. The authors state that HIV-1 infection of DCs does not lead to immune activation. However, no data are provided and the finding that p-Stat1 is phosphorylated upon HIV-1 incubation already suggests that some signaling and activation is occurring. Why is the activation so much faster (Fig 4A) compared to that observed with TLR8 ligands alone (Fig 5C). there are no data that HIV-1 indeed activates TLR8 and that TLR8 is involved in the observed pSTAT1.
Page 6. " Together, our results reveal that Snapin inhibits immune signaling through TLR8 in HIV-1 infected DCs." Is not supported by the data. Only stat1 phosphorylation is not enough to base this claim upon.

7. The authors state that HIV-1 targets snapin to prevent TLR8 activation and thereby DC activation. They show that activation of different signaling pathways is affected by snapin silencing (Fig 5). Interestingly, they already observe activation of these pathways in HIV-1 treated cells, suggesting that these cells become activated by HIV-1. They need to investigate cytokine production and DC maturation upon HIV-1 treatment to understand whether snapin silencing indeed allows DC maturation. It is not enough to investigate the activation of kinases and NF- κ B, as this already occurs in the non-silenced DCs with HIV-1

Is the effect of snapin on enhancing TLR8 activation only on TLR8 or are responses to other TLR ligands also enhanced?

Minor

8. in figure 3A, the authors show the effect of silencing different proteins on transmission to CD4+ T-cells. Why did the authors select snapin while DTNBP1 has a very similar effect on transmission?

9. In figure 3f it is unclear what each dot represents; a cell or a donor or experiment?

10. Of some of the experiments it is not stated how many times it was repeated / with how many donors

Referee #2:

Comments for the Authors

Elam Khatamzas et al. using phosphoproteomic approach found differentially phosphorylated proteins in DCs infected with HIV-1. Furthermore, a secondary siRNA screen, they identified signaling molecules involved in HIV-1 trans-infection of CD4 T-cells in DCs including a molecular motor protein snapin. Inhibition of snapin triggered a TLR8-mediated signaling induced by ssRNA40. The work could lead to better understanding of how HIV-1 infection avoid innate immune sensing in DCs and trans-infection of CD4 T-cells during antigen presentation.

Major points

1) In Fig. 1A, it would be better to use the same whole cell lysates across the board and probe them with phospho antibodies to get a better comparison rather than use the fractionated phosphoproteins. How come the beta-actin is induced in phospho-fraction with HIV-1? Why was 10 mins infection chosen? Is it the optimal response based on a time course? No mention of that.

2) In the DCs knockdown studies, as well as trans-infection studies, no mention whether mature or immature DCs were used. It is important because there is a big biologic difference between mature and immature DCs.

3) Fig. 3, even though confocal microscopy is a good way of showing protein association, it would give a lot of confidence if it would be verified by another way of showing association perhaps immunoprecipitation or FRET analysis. I know it is asking too much but it would very reassuring. Is snapin in phosphorylated or dephosphorylated state during association with HIV?

4) Where is Fig. 7?

Minor points

1) Page 8 line 7: what do you mean by "dint"?

Referee #3:

This interesting paper by Khatamzas et. al. investigates how HIV transcytoses myeloid dendritic cells (mDC) and trans-infects CD4 T cells without activating antiviral sensing by TLR8 in the mDC. The experiments are convincing and the topic important, but a few corrections are needed and additional information would enhance the paper.

1. Figure 1D in the legends describes a Venn diagram that is not present in the Figure and the text refers to the actual Figure 1D as 1E and to Figure 1E as Figure 1F. This need to be corrected. I suspect the Venn diagram was deleted and the Figure numbering in the text not renumbered following that.

2. I can't locate a methods section.
3. While STAT-1 phosphorylation is an excellent measure of Type I IFN signaling and other downstream molecules in the signaling pathway are assessed, it would be helpful to assess whether Type I IFN production is altered. Cytokines (IL-6 and TNF) are assessed with other stimuli but no assessment of cytokine production by mDC with HIV exposure after Snapin knockdown is performed. In addition to IL-6 and TNF, it would be great to see if there is an effect on IFN since that is the more relevant anti-viral cytokine.
4. While mutation of the dynein binding sites limits the effects of Snapin, it has not been shown that dynein binding plays a role so I might soften the sentence on page 5 stating that Snapin function in HIV-1 trafficking is "likely mediated by dynein binding" to state that it may require dynein binding.

1st Revision - authors' response

29 March 2017

Referee #1:

The authors have used an unbiased approach to identify signaling responses in dendritic cells upon HIV-1 binding. They show that there is a different phosphorylation profile in DCs treated with HIV-1 compared to uninfected controls. They have an unbiased method to identify which proteins are differentially phosphorylated upon HIV-1 uptake. They identify Snapin, as a protein that is somehow involved in HIV-1 transmission to CD4+ T-cells. The authors link the enhancing effect of snapin to TLR8 and hypothesize that HIV-1 uses snapin to escape TLR8 mediated antiviral immunity.

However, the main conclusions in the manuscript are not supported by the data, and some findings need to be elaborated. The authors state that Snapin downregulates TLR8 signaling in response to HIV-1. However, the data presented are not convincing as the authors fail to show TLR8 involvement in signaling upon HIV-1 entry. The role of Snapin in transmission remains unclear and more detailed experiments need to be performed to understand its role. The suggestion that Snapin is required for endosomal maturation would argue against a role in enhancing transmission as HIV-1 targeting to lysosomes will decrease virus transmission.

1. The authors suggest that Snapin as part of BLOC-1 complex is involved in HIV-1 transmission as silencing of Snapin decreased transmission. (Fig 3). Fig 3, only relative values are given, what is the amount of transmission (as in Fig 2, 8% transmission is low)? In Fig 3B there is hardly any virus taken up. It is difficult to assess whether this is indeed co-localization. The fact that Snapin is highly expressed in vesicles suggests that most endosomes contain Snapin, which explains the co-localisation but this does not mean that HIV-1 is specifically targeted to Snapin.

We thank all the reviewers for their helpful comments.

To assess the transfer for HIV-1 from monocyte derived DCs to T cells we co-cultured transduced DCs at a ratio of 1:1 with autologous CD4+ T cells activated with PHA. Transmission in these experiments was measured by the amount of p24 positive CD4+ T cells 72 hours after they have been co-incubated with HIV-exposed DCs. 8% transmission is what is to be expected utilizing primary dendritic cell and CD4+ T cell co-cultures. This level of HIV-1 transmission is comparable to publications in the field for example Menager and Littman, *Cell*, 164, 695-709, 2016, who used a similar high throughput screen and record similar levels of transmission. Relative values are given because infection levels differ between donors/experiments and the figure given shows the mean of these experiments. CD4+ T cells were tested for p24 expression at 72 hours to avoid measuring HIV-infected CD4+ cells that may have been infected via a cis pathway. At this time point there is no significant HIV-1 replication in DCs as measured by early expressed HIV-1 transcripts in DCs excluding a cis pathway in transmission here (Supplemental Figure 3). With regard to level of viral particles per cell observed in our experiments we used an MOI of 0.1-1 to infect DCs. It is expected for this MOI to generate generate few particles per cell as demonstrated by other groups undertaking similar methods (Coulon et al., *J Immunol*, 2016; Nikolic et al., *Blood*, 2011; Garcia et al., *Traffic et al.*, 2008). Many cells were imaged and quantified for each experiment, to increase the robustness of our analysis. We understand the concern raised by the referee about the possibility of inaccurately inferring colocalization

of an abundant protein with much less abundant viral particles. However, if the colocalization detected were a false positive result then we should not have been able to detect a difference in, for example Figure 3H. Figure 3H shows showing co-localization of abundantly expressed EEA1 with HIV-1 in the presence of WT Snapin but not dynein domain mutant Snapin.

2. HIV-1 uptake into Since the authors show that snapin enables the maturation of endosomes, and that snapin knockdown reduces HIV-1 transmission, it suggests that HIV-1 benefits from being taken up in late endosomes. This is not logical, as blocking endosomal maturation would prevent HIV-1 degradation. The authors need to show that the transmission is indeed enhanced because of snapin-induced endosomal maturation. As Snapin is also involved in secretion of vesicles, this might also be an important factor in facilitating transmission. Experiments need to be performed to further investigate this.

HIV-1 can be inhibited by destruction in lysosomes via autophagy (Ribiero et al., Nature 2016; Daussy et al., Oncotarget, 2015) but HIV-1 normally employs strategies to evade autophagic destruction in DCs (Blanchet et al., Immunity 2010). DCs take up HIV-1 particles into compartments where the virus particles are stable for up to 4 days (Geijtenbeek TB et al., Cell 2000; Yu HJ et al., Plos Pathogens 2008; Felts RL et al., PNAS 2010). These compartments are only mildly acidic, and possess tetraspanin markers such as but do not contain the early endosomal marker EEA1 or lysosomal LAMP-1 (Yu HJ et al., Plos Pathogens 2008, Garcia E et al., Traffic 2005). These compartments can be used for transfer of HIV-1 to T cells via virological synapses (Garcia E et al., Traffic 2005, McDonald D et al., Science 2003)

We did not show Snapin directs HIV-1 to an autophagosomal/lysosomal compartment but rather to Rab7 associated late endosomes. While these are expected to be more acidic than early endosomes there is no evidence blocking steps of endosomal maturation per se in DCs would prevent HIV-1 degradation without progression to lysosomes. We find Snapin is required to enhance HIV-1 transit to an innate sensing compartment hallmarked by expression of Rab7, as opposed to EEA1.

Snapin is known to act as a dynein adaptor and mediate synaptic vesicle pool size by late endosomal trafficking and endosomal sorting (Di Giovanni and Sheng., EMBO J, 2015). It is possible that Snapin might affect accumulation of HIV-1 within exocytic vesicles at the viral synapse in addition to diverting HIV-1 away from EEA1 positive innate sensing compartments. We agree it would be interesting to explore the role of Snapin in exocytosis of virions but this is beyond the scope of the current work.

3. What exactly is the mechanism of the enhanced transmission? The authors should elaborate on the effect of snapin silencing on infection of DCs. Show mechanism of enhanced transmission (is it cis/trans? Is uptake enhanced?) It is unclear in which step of transmission snapin is involved. On the transmission: please include controls that there is no cis-infection /transmission, for instance inhibitors.

Trans-infection is in these experiments the amount of p24 positive CD4+ T cells 72 hours after they have been co-incubated with HIV-exposed DCs. We chose this early time point to avoid measuring HIV-infected CD4+ cells that may have been infected via a cis pathway. We have included data to show that there is no effect of Snapin on early viral transcript expression within monocyte derived DCs (Supplemental data Figure 3) excluding enhanced cis-infection as a possibility for increased transmission in the effect.

4. Fig 3C. the transmission is very difficult to assess here, controls are missing (control-transfected DCs, as well as normal DCs) and absolute values need to be given. Figure 3C Why is there no reduction with the L99K mutant? (both are mutations in the dynein binding site)

We have included additional data with more replicate samples where significant reduction in transmission is achieved with the L99K mutant (Figure 3C). In the previous experiment we think the transfection levels were not high enough for an effect to be detected.

5. The colocalisation data on the confocal images are not convincing. Snapin silencing affects EEA1 and Rab7 expression, which can greatly affect co-localisation. More EEA1 expression can lead to more HIV-1 co-localisation and vice versa less Rab7 will lower the chance of colocalisation with HIV-1.

We include data to show that no statistical difference occurs in Rab7 expression following Snapin knockdown (Supplementary data Figure 4). As mentioned previously if the co-localization detected were a false positive result then we should not have been able to detect a difference in colocalization of EAA1 with HIV-1 in the presence of WT Snapin but not dynein domain mutant Snapin.

6. The authors state that HIV-1 infection of DCs does not lead to immune activation. However, no data are provided and the finding that p-Stat1 is phosphorylated upon HIV-1 incubation already suggests that some signaling and activation is occurring. Why is the activation so much faster (Fig 4A) compared to that observed with TLR8 ligands alone (Fig 5C). there are no data that HIV-1 indeed activates TLR8 and that TLR8 is involved in the observed pSTAT1.

Page 6. " Together, our results reveal that Snapin inhibits immune signaling through TLR8 in HIV-1 infected DCs." Is not supported by the data. Only stat1 phosphorylation is not enough to base this claim upon.

We have amended the text to clarify what was meant by the statement "HIV-1 infection of DCs does not lead to immune activation". Substantial literature supports the view that HIV-1 that has been internalized by immature DCs does not induce conventional DC activation and maturation as would be observed following a prototypic PRR stimulus (Borrow P., Curr Opin HIV AIDS; Sabado et al., Blood 2010). However, a small amount of signaling via TLR8 has been shown to occur that may activate low level transcriptional activity but not enough to induce robust proinflammatory cytokine production (Gringhuis et al., Nat Immunol, 2010). This would be consistent with what is observed at basal levels in DCs.

We included a time course of activation of pSTAT1 as phosphorylation and nuclear translocation of STAT1 occur at differing time points depending on the nature of PRR stimulus and cell type. R848 as an endosomal PRR where transfection of R848 in lyovec is required for the ligand to be internalised and sensed would be expected to take longer than stimulation of a surface TLR such as LPS. Detection of pSTAT1 following R848 triggering has previously been observed at 3.5 hr and 8 hr following R848 stimulation (Zannetti et al., J Biol Chem, 2010). Because of the differing nature of whole virus and single ligand it would not be expected for HIV-1 to induce identical signaling kinetics to R848 in terms of kinetics of transcriptional activation.

We have now included additional data to support our findings showing Snapin knockdown enhances HIV-1 mediated pro-inflammatory responses (Figure 4C).

7. The authors state that HIV-1 targets snapin to prevent TLR8 activation and thereby DC activation. They show that activation of different signaling pathways is affected by snapin silencing (Fig 5). Interestingly, they already observe activation of these pathways in HIV-1 treated cells, suggesting that these cells become activated by HIV-1. They need to investigate cytokine production and DC maturation upon HIV-1 treatment to understand whether snapin silencing indeed allows DC maturation. It is not enough to investigate the activation of kinases and NF-kB, as this already occurs in the non-silenced DCs with HIV-1. Is the effect of snapin on enhancing TLR8 activation only on TLR8 or are responses to other TLR ligands also enhanced?

We have provided additional data showing the effect of Snapin knockdown on release of cytokines from HIV-1 exposed DCs confirming induction of CXCL10, CXCL9 and TNF α in Snapin knockdown cells (Figure 4C). We have also provided additional data showing the expression of pro-inflammatory mediators is enhanced in Snapin knockdown cells infected with Sendai virus (Figure 5E).

Minor

8. in figure 3A, the authors show the effect of silencing different proteins on transmission to CD4+ T-cells. Why did the authors select snapin while DTNBP1 has a very similar effect on transmission?

We selected Snapin rather than DTNBP1 to pursue as overall Snapin had the greatest effect on HIV-1 transmission. As DTNBP1 binds to Snapin it is likely it will operate in the same pathway utilised by HIV-1 however dissecting the specific function of more than one protein in HIV-1 transmission is beyond the scope of the current work.

9. In figure 3f it is unclear what each dot represents; a cell or a donor or experiment?

In the graph of figure 3f each dot represents a cell. The graph shows data from one representative experiment. The experiment was repeated 3 times.

10. Of some of the experiments it is not stated how many times it was repeated / with how many donors.

We have now clarified in the figure legends how many times each experiment was repeated and with how many donors.

Referee #2

Elham Khatamzas et al. using phosphoproteomic approach found differentially phosphorylated proteins in DCs infected with HIV-1. Furthermore, a secondary siRNA screen, they identified signaling molecules involved in HIV-1 trans-infection of CD4 T-cells in DCs including a molecular motor protein snapin. Inhibition of snapin triggered a TLR8-mediated signaling induced by ssRNA40. The work could lead to better understanding of how HIV-1 infection avoid innate immune sensing in DCs and trans-infection of CD4 T-cells during antigen presentation.

Major points

1) In Fig. 1A, it would be better to use the same whole cell lysates across the board and probe them with phospho antibodies to get a better comparison rather than use the fractionated phosphoproteins. How come the beta-actin is induced in phospho-fraction with HIV-1? Why was 10 mins infection chosen? Is it the optimal response based on a time course? No mention of that.

Figure 1A does show whole cell lysates from the same samples immunoblotted using antiphosphotyrosine antibody on DC whole cell lysates (WCL) in the presence and absence of HIV-1 for 10 mins (lane 1 and 2), the same WCL diluted 1:10 (lanes 3 and 4), and phosphoenriched fractions derived from the same WCL (lanes 5 and 6). Actin is depleted in the phospho-fractions compared to the matched WCL sample from which the phospho-fraction is derived. Anti-actin was used as a loading control for the WCL. 10 minutes was chosen as this was the optimal time for detection of differences in phosphorylation by immunoblot following exposure of DCs to HIV-1. We have now stipulated this in the text.

2) In the DCs knockdown studies, as well as trans-infection studies, no mention whether mature or immature DCs were used. It is important because there is a big biologic difference between mature and immature DCs.

We have now amended the text to stipulate that immature DCs were used in the experiments.

3) Fig. 3, even though confocal microscopy is a good way of showing protein association, it would give a lot of confidence if it would be verified by another way of showing association perhaps immunoprecipitation or FRET analysis. I know it is asking too much but it would very reassuring. Is snapin in phosphorylated or dephosphorylated state during association with HIV?

It was difficult to show interaction of HIV-1 with Snapin by methods other than co-localization for this study. Unfortunately, the anti-Snapin antibodies available do not work well for immunoprecipitation and immunoblot in DCs. Labelling HIV-1 particles with FRET donors or acceptors would most likely affect viral structure too much to then facilitate binding to respective proteins.

4) Where is Fig. 7?

We have corrected the text and removed the incorrect reference to Figure 7.

Minor points

1) Page 8 line 7: what do you mean by "dint"?

We have changed the text to read "It is possible HIV-1, by binding to specific tolerogenic entry receptors such as DC-SIGN, activates PKA to phosphorylate Snapin and secure Snapin binding to

the SNARE complex”.

Referee #3

This interesting paper by Khatamzas et. al. investigates how HIV transcytoses myeloid dendritic cells (mDC) and trans-infects CD4 T cells without activating antiviral sensing by TLR8 in the mDC. The experiments are convincing and the topic important, but a few corrections are needed and additional information would enhance the paper.

1. Figure 1D in the legends describes a Venn diagram that is not present in the Figure and the text refers to the actual Figure 1D as 1E and to Figure 1E as Figure 1F. This need to be corrected. I suspect the Venn diagram was deleted and the Figure numbering in the text not renumbered following that.

We have corrected the Figure labelling in the legends and associated text for figure 1 to correctly label Figure 1D and 1E.

2. I can't locate a methods section.

The methods section is now included with the text.

3. While STAT-1 phosphorylation is an excellent measure of Type I IFN signaling and other downstream molecules in the signaling pathway are assessed, it would be helpful to assess whether Type I IFN production is altered. Cytokines (IL-6 and TNF) are assessed with other stimuli but no assessment of cytokine production by mDC with HIV exposure after Snapin knockdown is performed. In addition to IL-6 and TNF, it would be great to see if there is an effect on IFN since that is the more relevant anti-viral cytokine.

We have included a figure showing the effect of Snapin knockdown on expression of CXCL10 and CXCL9 and pro-inflammatory mediator TNF α by HIV-1 (Figure 4C). We have been unable to show an effect of HIV-1 in induction of interferon likely as monocyte derived DCs are poor inducers of type 1 interferon generally.

4. While mutation of the dynein binding sites limits the effects of Snapin, it has not been shown that dynein binding plays a role so I might soften the sentence on page 5 stating that Snapin function in HIV-1 trafficking is "likely mediated by dynein binding" to state that it may require dynein binding.

We have changed the text on page 5 as recommended by reviewer 3 to say Snapin function in HIV-1 trafficking “may be mediated by dynein binding” rather than “likely mediated”.

References

- Ménager MM, Littman DR. Actin Dynamics Regulates Dendritic Cell-Mediated Transfer of HIV-1 to T Cells. *Cell*. 2016 Feb 11;164(4):695-709. doi: 10.1016/j.cell.2015.12.036.
- Coulon PG, Richetta C, Rouers A, Blanchet FP, Urrutia A, Guerbois M, Piguet V, Theodorou I, Bet A, Schwartz O, Tangy F, Graff-Dubois S, Cardinaud S, Moris A. HIV-Infected Dendritic Cells Present Endogenous MHC Class II-Restricted Antigens to HIV-Specific CD4+ T Cells. *J Immunol*. 2016 Jul 15;197(2):517-32. doi: 10.4049/jimmunol.1600286.
- HIV-1 activates Cdc42 and induces membrane extensions in immature dendritic cells to facilitate cell-to-cell virus propagation. Nikolic DS, Lehmann M, Felts R, Garcia E, Blanchet FP, Subramaniam S, Piguet V. *Blood*. 2011 Nov 3;118(18):4841-52. doi: 10.1182/blood-2010-09-305417.
- Garcia E, Nikolic DS, Piguet V. HIV-1 replication in dendritic cells occurs through a tetraspanin-containing compartment enriched in AP-3. *Traffic*. 2008 Feb;9(2):200-14.

Gringhuis SI, van der Vlist M, van den Berg LM, den Dunnen J, Litjens M, Geijtenbeek TB. HIV-1 exploits innate signaling by TLR8 and DC-SIGN for productive infection of dendritic cells. *Nat Immunol*. 2010 May;11(5):419-26. doi: 10.1038/ni.1858.

Coulon PG, Richetta C, Rouers A, Blanchet FP, Urrutia A, Guerbois M, Piguet V, Theodorou I, Bet A, Schwartz O, Tangy F, Graff-Dubois S, Cardinaud S, Moris A. HIV-Infected Dendritic Cells Present Endogenous MHC Class II-Restricted Antigens to HIV-Specific CD4⁺ T Cells. *J Immunol*. 2016 Jul 15;197(2):517-32. doi: 10.4049/jimmunol.1600286.

HIV-1 activates Cdc42 and induces membrane extensions in immature dendritic cells to facilitate cell-to-cell virus propagation. Nikolic DS, Lehmann M, Felts R, Garcia E, Blanchet FP, Subramaniam S, Piguet V. *Blood*. 2011 Nov 3;118(18):4841-52. doi: 10.1182/blood-2010-09-305417.

Garcia E, Nikolic DS, Piguet V. HIV-1 replication in dendritic cells occurs through a tetraspanin-containing compartment enriched in AP-3. *Traffic*. 2008 Feb;9(2):200-14.

Zannetti C, Bonnay F, Takeshita F, Parroche P, Ménétrier-Caux C, Tommasino M, Hasan UA. C/EBP δ and STAT-1 are required for TLR8 transcriptional activity. *J Biol Chem*. 2010 Nov 5;285(45):34773-80. doi: 10.1074/jbc.M110.133884.

Borrow, P. Innate immunity in acute HIV-1 infection. *Curr Opin HIV AIDS* 6, 353-363 (2011).

Sabado, R.L. *et al.* Evidence of dysregulation of dendritic cells in primary HIV infection. *Blood* 116, 3839-3852 (2010)

Gringhuis SI, Hertoghs N, Kaptein TM, Zijlstra-Willems EM, Sarrami-Forooshani R, Sprokholt JK, van Teijlingen NH, Kootstra NA, Booiman T, van Dort KA, Ribeiro CM, Drewniak A, Geijtenbeek TB. HIV1 blocks the signaling adaptor MAVS to evade antiviral host defense after sensing of abortive HIV-1 RNA by the host helicase DDX3. *Nat Immunol*. 2017 Feb;18(2):225-235. doi: 10.1038/ni.3647.

Manel N, Hogstad B, Wang Y, Levy DE, Unutmaz D, Littman DR. A cryptic sensor for HIV-1 activates antiviral innate immunity in dendritic cells. *Nature*. 2010 Sep 9;467(7312):214-7. doi: 10.1038/nature09337. Erratum in: *Nature*. 2011 Feb 17;470(7334):424. PMID: 20829794

Ribeiro CM, Sarrami-Forooshani R, Setiawan LC, Zijlstra-Willems EM, van Hamme JL, Tigchelaar W, van der Wel NN, Kootstra NA, Gringhuis SI, Geijtenbeek TB. Receptor usage dictates HIV-1 restriction by human TRIM5 α in dendritic cell subsets. *Nature*. 2016 Dec 15;540(7633):448-452. doi: 10.1038/nature20567.

Daussy CF, Beaumelle B, Espert L. Autophagy restricts HIV-1 infection. *Oncotarget*. 2015 Aug 28;6(25):20752-3. No abstract available.

Blanchet FP, Moris A, Nikolic DS, Lehmann M, Cardinaud S, Stalder R, Garcia E, Dinkins C, Leuba F, Wu L, Schwartz O, Deretic V, Piguet V. Human immunodeficiency virus-1 inhibition of immunoamphisomes in dendritic cells impairs early innate and adaptive immune responses. *Immunity*. 2010 May 28;32(5):654-69. doi: 10.1016/j.immuni.2010.04.01

Di Giovanni J, Sheng ZH. Regulation of synaptic activity by snapin-mediated endolysosomal transport and sorting. *EMBO J*. 2015 Aug 4;34(15):2059-77. doi: 10.15252/embj.201591125.

Eitan E, Suire C, Zhang S, Mattson MP. Impact of lysosome status on extracellular vesicle content and release. *Ageing Res Rev*. 2016 Dec;32:65-74. doi: 10.1016/j.arr.2016.05.001. Review.

Geijtenbeek TB, Torensma R, van Vliet SJ, van Duijnhoven GC, Adema GJ, van Kooyk Y, Figdor CG. Identification of DC-SIGN, a novel dendritic cell-specific ICAM-3 receptor that supports primary immune responses. *Cell*. 2000 Mar 3;100(5):575-85.

Yu HJ, Reuter MA, McDonald D. HIV traffics through a specialized, surface-accessible intracellular

compartment during trans-infection of T cells by mature dendritic cells. *PLoS Pathog.* 2008 Aug 22;4(8):e1000134. doi: 10.1371/journal.ppat.1000134.

Felts RL, Narayan K, Estes JD, Shi D, Trubey CM, Fu J, Hartnell LM, Ruthel GT, Schneider DK, Nagashima K, Bess JW Jr, Bavari S, Lowekamp BC, Bliss D, Lifson JD, Subramaniam S. 3D visualization of HIV transfer at the virological synapse between dendritic cells and T cells. *Proc Natl Acad Sci U S A.* 2010 Jul 27;107(30):13336-41. doi: 10.1073/pnas.1003040107.

Garcia E, Pion M, Pelchen-Matthews A, Collinson L, Arrighi JF, Blot G, Leuba F, Escola JM, Demaurex N, Marsh M, Piguet V. HIV-1 trafficking to the dendritic cell-T-cell infectious synapse uses a pathway of tetraspanin sorting to the immunological synapse. *Traffic.* 2005 Jun;6(6):488-501.

McDonald D, Wu L, Bohks SM, KewalRamani VN, Unutmaz D, Hope TJ. Recruitment of HIV and its receptors to dendritic cell-T cell junctions. *Science.* 2003 May 23;300(5623):1295-7.

2nd Editorial Decision

15 May 2017

Thanks for submitting your revised manuscript to The EMBO Journal. Your study has now been re-reviewed by the three referees.

The referees appreciate that the analysis has been strengthened. However, there are still some issues that have to be sorted out before acceptance here:

Ref #1 (point 1) and referee #2 both raise issues with the co-localization data: I would suggest that you provide a balanced discussion of the data including discussing alternative mechanisms.

Ref #1 (point 2): Please carry out the suggested experiment to look at T cell infection.

Ref #1 (minor comment): Would be great to include the TLR8 silencing experiment

So we are almost there! Resolving these last issues will strengthen the analysis and tie everything nicely together.

Let me know if we need to discuss anything further

REFEREE REPORTS

Referee #1:

The authors have addressed some of the concerns, especially with regard to the increased immunoactivation upon Snapin knockdown, which is now clear. Some concerns remain.

Co-localization with EEA1: The authors state that Snapin diverts HIV-1 away from EEA1 positive compartments but the data remain unclear, especially the confocal images. There seems to be a higher expression of EEA1 after Snapin knockdown, which can cause more co-localization as the co-localization is not really clearly observed. I understand that the authors calculate the Li's coefficient but is this not affected by a stronger EEA1 staining? As the authors state, there is co-localization after transfection of cells with Snapin mutants but the vesicles seem different (caused by overexpression Snapin mutants?) and the concern regarding increased EEA1 and weak co-localization with HIV-1 remain.

Relocalization or defect in secretion/synapse: The authors conclude from these co-localization studies that Snapin enhances HIV-1 transmission because it diverts the virus away from EEA1 positive compartments, but they completely disregard an effect of Snapin on secretion of the virus/formation of synaptic vesicles. Their rebuttal is not sufficient. The data so far presented only show a decrease in HIV-1 transmission after Snapin knockdown but this could be due to either diversion of virus in other vesicles (the paper's hypothesis) or decreased secretion (synapse

formation). This needs to be clarified. It is an important issue as the authors state that transmission is decreased because of the changes in trafficking from early to late endosomes but it might just as well be from secretion/synapse formation.

Furthermore, infection by p24 production is not a very specific method and p24 staining by flow cytometry should show whether T cells are infected. The suppl data fig 3 only show that early transcription is not affected by Snapin but the question was whether the decrease in transmission is not caused by less DC infection and not less T cell infection.

Minor comment.

The authors state very strongly that Snapin prevents TLR8 triggering by HIV-1 (page 6, middle: Together, our results reveal that Snapin inhibits immune signaling through TLR8 in HIV-1 infected DCs.) but there is no evidence that HIV-1 in their model triggers TLR8. At the very least TLR8 knockdown should show that the responses are caused by TLR8.

Referee #2:

The significance of the study is the identification of signaling molecule snapin involved in infection of CD4 T cells mediated by HIV-infected mDCs that avoids innate immune sensing. This work is important to better understand how HIV exploit the host in avoiding innate immune sensing by pattern recognition receptors to promote transmission. Even though the co-localization data were not verified by other association experiments, the work was strengthened by additional data that directly or indirectly support the conclusions. My concerns were addressed by the authors and I have no further additional suggestions.

Referee #3:

My concerns were addressed fully.

2nd Revision - authors' response

20 July 2017

Referee #1:

The authors have addressed some of the concerns, especially with regard to the increased immunoactivation upon Snapin knockdown, which is now clear. Some concerns remain.

Co-localization with EEA1: The authors state that Snapin diverts HIV-1 away from EEA1 positive compartments but the data remain unclear, especially the confocal images. There seems to be a higher expression of EEA1 after Snapin knockdown, which can cause more co-localization as the co-localization is not really clearly observed. I understand that the authors calculate the Li's coefficient but is this not affected by a stronger EEA1 staining? As the authors state, there is co-localization after transfection of cells with Snapin mutants but the vesicles seem different (caused by overexpression Snapin mutants?) and the concern regarding increased EEA1 and weak colocalization with HIV-1 remain.

Relocalization or defect in secretion/synapse: The authors conclude from these co-localization studies that Snapin enhances HIV-1 transmission because it diverts the virus away from EEA1 positive compartments, but they completely disregard an effect of Snapin on secretion of the virus/formation of synaptic vesicles. Their rebuttal is not sufficient. The data so far presented only show a decrease in HIV-1 transmission after Snapin knockdown but this could be due to either diversion of virus in other vesicles (the paper's hypothesis) or decreased secretion (synapse formation). This needs to be clarified. It is an important issue as the authors state that transmission is decreased because of the changes in trafficking from early to late endosomes but it might just as well be from secretion/synapse formation.

Furthermore, infection by p24 production is not a very specific method and p24 staining by flow cytometry should show whether T cells are infected. The suppl data fig 3 only show that early

transcription is not affected by Snapin but the question was whether the decrease in transmission is not caused by less DC infection and not less T cell infection.

Minor comment.

The authors state very strongly that Snapin prevents TLR8 triggering by HIV-1 (page 6, middle: Together, our results reveal that Snapin inhibits immune signaling through TLR8 in HIV-1 infected DCs.) but there is no evidence that HIV-1 in their model triggers TLR8. At the very least TLR8 knockdown should show that the responses are caused by TLR8.

Ref 1 (point 1):

We have changed the text describing the confocal on page 6 to include:

In addition, following Snapin knockdown there appeared to be a different morphological appearance of EEA1 and Rab7 vesicles within DCs however the overall intensity of EEA1 or Rab7 staining did not change (**Appendix Fig S4**).

And we have provided a balanced discussion of these findings in the discussion page 8:

In this role, Snapin inhibits HIV-1 detection by TLR8 in endosomes, and subsequently promotes *trans*-infection and viral spread through the host. Although this study focused on defining a role for Snapin in endosomal maturation and TLR signaling it is also possible Snapin may be involved in secretion of HIV-1 from DCs across the viral synapse akin to its previously described role at neurological synapses (*Buxton, Zhang et al., 2003, Ilardi, Mochida et al., 1999*). While we observed enhanced co-localization of HIV-1 with EEA1 positive vesicles following depletion of Snapin, the morphological appearance of EEA1 appeared to subtly change between Ctrl and Snapin knockdown cells. It is possible as part of the BLOC1 complex Snapin knockdown leads to a failure of fragmentation or enhanced fusion of early endosomes in addition to its role in TLR8 sensing.

Ref 1 (point 2):

We have included data demonstrating that Snapin knockdown has no significant effect on internalization of HIV-1 into DCs as measured by ATTO488 HIV-1 staining by FACS. Here DCs were transfected with Ctrl and Snapin shRNAs for 48 hr prior to exposure to HIV-1 for 2 hr (**Appendix Fig S2**). This combined with the qPCR for early expressed transcripts (**Appendix Fig S5**) suggests the reduced p24 detected in CD4⁺ T cells derived from DC CD4⁺ T cell co-cultures occurs as a result of increased *trans*-infection.

Ref 1 (minor comment):

We have included the TLR8 silencing experiment (**Appendix Fig S6**) and changed the associated text on page 6 to: Furthermore depletion of TLR8 in Snapin knockdown cells impaired the enhanced release of TNF α (**Appendix Fig S6**) suggesting that Snapin inhibits immune signaling through TLR8 in HIV-1 infected DCs.

3rd Editorial Decision

26 July 2017

Thanks for submitting your revised manuscript to The EMBO Journal. Your revision has now been seen by referee #1 and the comments are provided below.

The referee appreciates the introduced revisions and support publication here. Please respond to the issue of using THP1 cells in the point-by-point response. If appropriate you can also discuss this issue in the manuscript, but I will leave that up to you.

So the manuscript is ready to be accepted. Congratulations!

REFEREE REPORT

Referee #1:

The authors have addressed most of the comments.

It is unclear why the authors decided to silence TLR8 in THP1 cells but not in the relevant dendritic cells, which they use for their immune responses. THP-1 cells do not express DC-SIGN that has been shown to be involved in HIV-1 signalling by the authors in a previous study. Nevertheless these data suggest that TLR8 is involved in HIV-1 induced immune responses upon TLR8 silencing assuming that THP-1 is similar to DCs.

Alison Simmons
EMBO Journal
EMBOJ-2016-95364R2